# *Drosophila* Synaptotagmin 7 negatively regulates synaptic vesicle release and replenishment in a dosage-dependent manner

Zhuo Guan[†], Monica C Quiñones-Frías[†], Yulia Akbergenova, J Troy Littleton*

The Picower Institute for Learning and Memory, Department of Biology and Department of Brain and Cognitive Sciences, Massachusetts Institute of Technology, Cambridge, United States

**Abstract** Synchronous neurotransmitter release is triggered by $Ca^{2+}$ binding to the synaptic vesicle protein Synaptotagmin 1, while asynchronous fusion and short-term facilitation is hypothesized to be mediated by plasma membrane-localized Synaptotagmin 7 (SYT7). We generated mutations in *Drosophila Syt7* to determine if it plays a conserved role as the $Ca^{2+}$ sensor for these processes. Electrophysiology and quantal imaging revealed evoked release was elevated 2-fold. *Syt7* mutants also had a larger pool of readily-releasable vesicles, faster recovery following stimulation, and intact facilitation. *Syt1/Syt7* double mutants displayed more release than *Syt1* mutants alone, indicating SYT7 does not mediate the residual asynchronous release remaining in the absence of SYT1. SYT7 localizes to an internal membrane tubular network within the peri-active zone, but does not enrich at active zones. These findings indicate the two $Ca^{2+}$ sensor model of SYT1 and SYT7 mediating all phases of neurotransmitter release and facilitation is not applicable at *Drosophila* synapses.

**\*For correspondence:**
troy@mit.edu

[†]These authors contributed equally to this work

**Competing interests:** The authors declare that no competing interests exist.

## Introduction

Neurotransmitter release from presynaptic terminals is the primary mechanism of synaptic communication and is mediated by fusion of synaptic vesicles (SVs) with the plasma membrane at specific sites known as active zones (AZs) (*Katz, 1969*; *Südhof, 2013*; *Zhai and Bellen, 2004*). A highly conserved protein machinery composed of the SNARE complex drives fusion between the SV and AZ lipid bilayers (*Littleton et al., 1998*; *Söllner et al., 1993*; *Sutton et al., 1998*; *Tucker et al., 2004*). $Ca^{2+}$ influx through voltage-gated $Ca^{2+}$ channels functions as the trigger to activate the fusion process (*Borst and Sakmann, 1996*; *Katz and Miledi, 1970*; *Katz and Miledi, 1967*; *Schneggenburger and Rosenmund, 2015*; *Südhof, 2012*). The majority of SVs fuse during a synchronous phase that occurs within a few milliseconds of $Ca^{2+}$ entry (*Borst and Sakmann, 1996*; *Goda and Stevens, 1994*; *Llinás et al., 1981*; *Sabatini and Regehr, 1996*; *Yoshihara and Littleton, 2002*). Many synapses also have an asynchronous component that results in SV release over hundreds of milliseconds (*Goda and Stevens, 1994*; *Hefft and Jonas, 2005*; *Kaeser and Regehr, 2014*; *Yoshihara and Littleton, 2002*). Asynchronous release normally accounts for less than 5% of SV fusion following single action potentials at *Drosophila* neuromuscular junctions (NMJs) (*Jorquera et al., 2012*). This slower phase of release becomes more prominent during high rates of stimulation (*Atluri and Regehr, 1998*; *Lu and Trussell, 2000*; *Rozov et al., 2019*; *Zucker and Regehr, 2002*) and mediates all SV fusion at some neuronal connections (*Best and Regehr, 2009*; *Peters et al., 2010*). Changes in the kinetics and amount of SV fusion also occur during high frequency stimulation, resulting in facilitation or depression depending on the synapse (*Zucker and*

*Regehr, 2002*). Defining the molecular machinery and Ca$^{2+}$ sensors that regulate the distinct modes and kinetics of SV release is essential for understanding synaptic transmission.

The Synaptotagmin (SYT) family of Ca$^{2+}$ binding proteins contain key regulators that control the timing of SV release. SYT proteins have a transmembrane domain and two Ca$^{2+}$ binding C2 domains termed C2A and C2B (*Adolfsen et al., 2004*; *Adolfsen and Littleton, 2001*; *Perin et al., 1990*; *Sugita et al., 2002*; *Ullrich and Südhof, 1995*). Mammals have three SYT family members that localize to SVs (SYT1, SYT2 and SYT9), while *Drosophila* contains a single member of the SV subfamily (SYT1) (*Littleton et al., 1993a*; *Pang et al., 2006*; *Xu et al., 2007*). These SYT isoforms bind Ca$^{2+}$ and activate synchronous fusion of SVs via interactions with membranes and the SNARE complex (*Chang et al., 2018*; *Chapman and Jahn, 1994*; *Fernández-Chacón et al., 2001*; *Geppert et al., 1994*; *Guan et al., 2017*; *Lee et al., 2013*; *Lee and Littleton, 2015*; *Littleton et al., 1994*; *Littleton et al., 1993b*; *Mackler et al., 2002*; *Nishiki and Augustine, 2004*; *Tucker et al., 2004*; *Xu et al., 2007*; *Yoshihara and Littleton, 2002*; *Young and Neher, 2009*). Beyond SV localized SYTs, SYT7 is the only other family member implicated in Ca$^{2+}$-dependent SV trafficking, although additional SYT isoforms participate in Ca$^{2+}$-dependent fusion of other secretory organelles and dense core vesicles (DCVs) (*Adolfsen et al., 2004*; *Cao et al., 2011*; *Dean et al., 2012*; *Li et al., 1995*; *Moghadam and Jackson, 2013*; *Park et al., 2014*; *Shin et al., 2002*; *Yoshihara et al., 2005*).

Multiple mechanisms have been proposed to mediate the asynchronous component of neurotransmitter release, including distinct Ca$^{2+}$ sensors, heterogeneity in SV protein content, SV distance from Ca$^{2+}$ channels, distinct Ca$^{2+}$ entry pathways, or regulation of Ca$^{2+}$ extrusion and buffering (*Chanaday and Kavalali, 2018*; *Fesce, 1999*; *Kaeser and Regehr, 2014*; *Pang and Südhof, 2010*; *Rozov et al., 2019*; *Zucker and Regehr, 2002*). Although several mechanisms may contribute, the observation that *Syt1* mutants have enhanced asynchronous release indicates another Ca$^{2+}$ sensor(s) activates the remaining slower Ca$^{2+}$-dependent component of exocytosis (*Huson et al., 2019*; *Kochubey and Schneggenburger, 2011*; *Nishiki and Augustine, 2004*; *Turecek and Regehr, 2019*; *Yang et al., 2010*; *Yoshihara et al., 2010*; *Yoshihara and Littleton, 2002*). SYT7 has emerged as a popular candidate for the asynchronous Ca$^{2+}$ sensor (*Bacaj et al., 2013*; *Chen et al., 2017*; *Maximov et al., 2008*; *Turecek and Regehr, 2019*; *Turecek and Regehr, 2018*; *Weber et al., 2014*; *Wen et al., 2010*). SYT7 has also been postulated to function as the Ca$^{2+}$ sensor for short-term synaptic facilitation (*Chen et al., 2017*; *Jackman et al., 2016*; *Turecek and Regehr, 2018*). SYT7 has higher Ca$^{2+}$ sensitivity, tighter membrane-binding affinity and longer Ca$^{2+}$-lipid disassembly kinetics than SYT1 (*Hui et al., 2005*; *Sugita et al., 2002*; *Sugita et al., 2001*; *Voleti et al., 2017*). These properties suggest SYT7 may regulate SV dynamics farther away from the AZ Ca$^{2+}$ nanodomains that are required for SYT1 activation, or during temporal windows following the decay of the initial peak of Ca$^{2+}$ influx. Together, these data have led to a two Ca$^{2+}$ sensor model for evoked SV exocytosis, with SYT1 triggering the rapid synchronous phase of neurotransmitter release and SYT7 mediating asynchronous fusion and facilitation.

Although SYT7 manipulations can alter asynchronous release and facilitation at some synapses, several studies have suggested alternative explanations or identified unrelated defects in SV trafficking (*Figure 1A*). The recent observation that asynchronous release at mammalian synapses is anticorrelated with the levels of the synchronous Ca$^{2+}$ sensors SYT1 and SYT2, but does not correlate with SYT7, prompted re-interpretation of earlier data on the protein's function (*Turecek and Regehr, 2019*). Besides asynchronous release and facilitation, mammalian SYT7 has been implicated in SV endocytosis, SV replenishment, SV pool mobility, and DCV fusion and replenishment (*Bacaj et al., 2015*; *Dolai et al., 2016*; *Durán et al., 2018*; *Fukuda et al., 2004*; *Gustavsson et al., 2011*; *Li et al., 2017*; *Liu et al., 2014*; *Schonn et al., 2008*; *Tsuboi and Fukuda, 2007*; *Virmani, 2003*; *Wu et al., 2015*). SYT7 has also been shown to regulate cell migration, lysosomal fusion and membrane repair in non-neuronal cells (*Barzilai-Tutsch et al., 2018*; *Chakrabarti et al., 2003*; *Colvin et al., 2010*; *Czibener et al., 2006*; *Flannery et al., 2010*; *Jaiswal et al., 2004*; *Martinez et al., 2000*; *Reddy et al., 2001*; *Zhao et al., 2008*).

Similar to the uncertainty surrounding SYT7 function, its subcellular localization is also unclear, with different studies localizing the protein to the plasma membrane, DCVs, lysosomes, endosomes or other internal compartments (*Adolfsen et al., 2004*; *Czibener et al., 2006*; *Flannery et al., 2010*; *Martinez et al., 2000*; *Mendez et al., 2011*; *Monterrat et al., 2007*; *Schonn et al., 2008*; *Shin et al., 2002*; *Sugita et al., 2001*; *Zhao et al., 2008*). A key supporting argument for SYT7 as the asynchronous Ca$^{2+}$ sensor is its reported localization to the AZ plasma membrane, positioning it

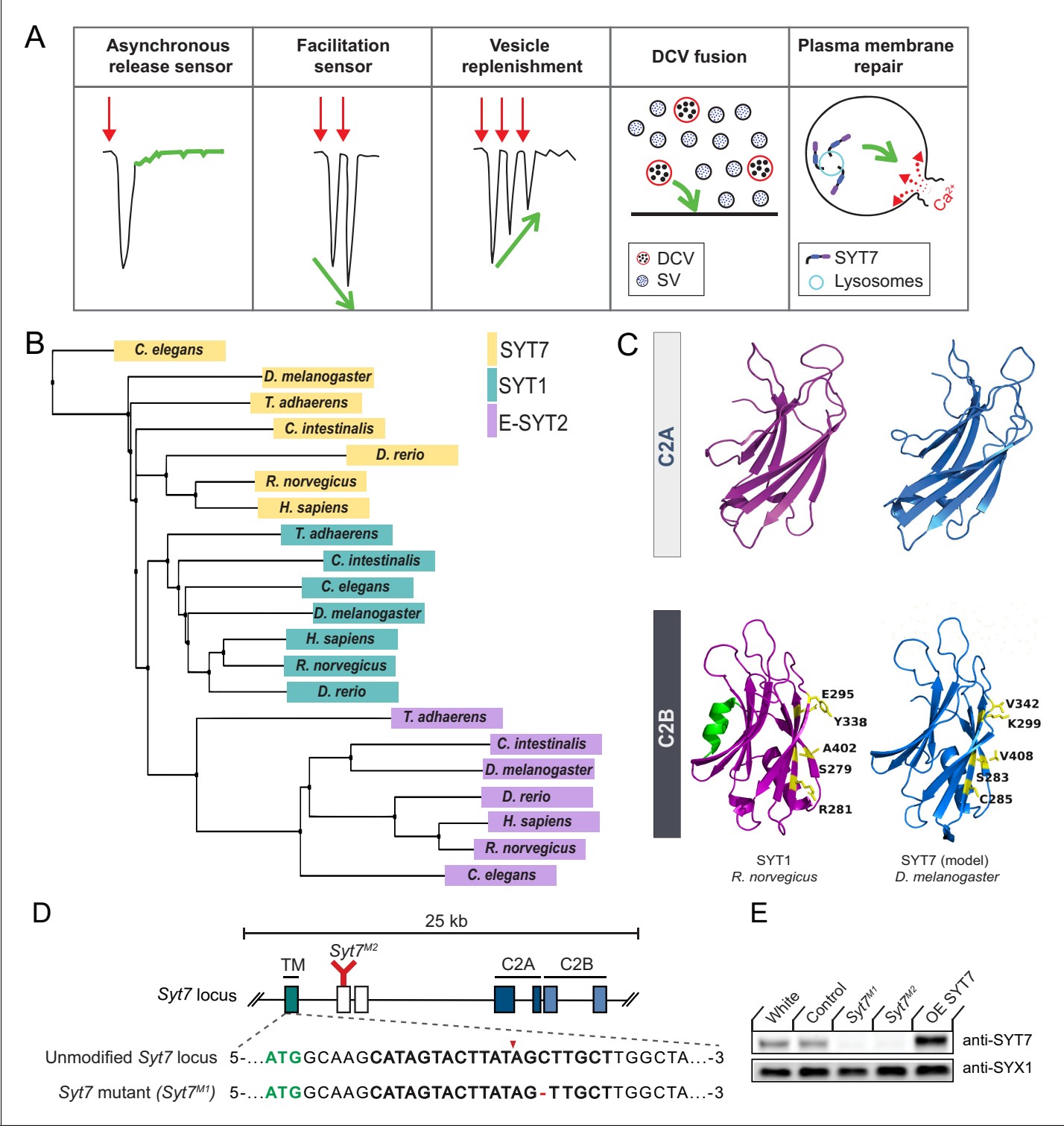

**Figure 1.** SYT1 and SYT7 comparison and generation of *Syt7* mutants. (**A**) Proposed roles for SYT7 in Ca²⁺-regulated membrane trafficking. (**B**) Phylogenetic tree of SYT1, SYT7 and E-SYT2 from the indicated species generated using the BLOSUM62 matrix with neighbor joining clustering. (**C**) Comparison of the structure of the C2A and C2B domains of *R. norvegicus* SYT1 (magenta) with a homology model of *D. melanogaster* SYT7 (blue). The C2B residues that form the SYT1-SNARE complex primary binding site are highlighted in yellow, with the counterpart changes noted in SYT7. The C2B HB helix in SYT1 is highlighted in green and missing from SYT7. (**D**) Diagram of the *Syt7* genomic locus on chromosome four with coding exons indicated with boxes. Exon 1 (teal) encodes the intravesicular and transmembrane (TM) domains; exons 2 and 3 (white) encode the linker region; exons 4 and 5 encode the C2A domain (dark blue); and exons 6 and 7 encode the C2B domain (light blue). The location of the *Syt7ᴹ²* Minos transposon

*Figure 1 continued on next page*

*Figure 1 continued*

insertion in exon two is indicated in red. Sequence of the *Syt7^M1* CRISPR mutant versus control is shown below with the start codon in green. The guide RNA sequence used to target *Syt7* is bolded, with the cleavage site noted by the red arrowhead and the deleted cytosine with a red dash. (E) Western blot of SYT7 protein levels in head extracts of *white*, CRISPR control, *Syt7^M1*, *Syt7^M2* and *elav^C155*-GAL4; UAS-*Syt7* (OE SYT7) with anti-SYT7 antisera (top panel). Syntaxin 1 (SYX1) antisera was used as a loading control (bottom panel). SYT7 is overexpressed 2.48 ± 0.4 fold compared to controls (p<0.05, Mann-Whitney unpaired t-test, n = 4).

The online version of this article includes the following figure supplement(s) for figure 1:

**Figure supplement 1.** SYT1 and SYT7 sequence comparisons.

at sites of SV fusion (*Sugita et al., 2001*). If SYT7 were present on endosomes or other internal membrane compartments, it would be more compatible with a role in SV trafficking rather than the fusion process itself. In summary, conflicting studies have generated confusion over how SYT7 contributes to neurotransmission and if the protein plays distinct roles across different neuronal subpopulations or species.

To examine the function of SYT7 in *Drosophila*, we generated and characterized *Syt7* null mutants. The *Drosophila* NMJ exhibits similar asynchronous release and facilitation properties to those of mammals (*Jan and Jan, 1976*; *Jorquera et al., 2012*; *Yoshihara and Littleton, 2002*), making it a useful system to examine evolutionary conserved functions of SYT7 in neurotransmitter release. We found *Syt7* mutants and *Syt1; Syt7* double mutants display increased evoked neurotransmitter release, indicating SYT7 negatively regulates SV fusion independent of SYT1. In addition, CRISPR-mediated tagging of the endogenous *Syt7* locus indicates SYT7 localizes to a tubular network inside the presynaptic terminal that resides within the peri-active zone (peri-AZ) region, but is not enriched at sites of SV fusion. These data define a role for SYT7 in restricting SV availability and release, and indicate SYT7 is not a major Ca$^{2+}$ sensor for asynchronous fusion and facilitation in *Drosophila*.

## Results

### Evolutionary conservation and structural comparison of SYT1 and SYT7

Synaptotagmins form one of the largest protein families involved in membrane tracking, with 17 *Syt* genes encoded in mammals and 7 *Syt* genes found in *Drosophila* (*Adolfsen and Littleton, 2001*; *Craxton, 2010*; *Sugita et al., 2002*). Unlike the SV subfamily of SYTs, only a single *Syt7* gene is present in vertebrate and invertebrate genomes, making phenotypic comparisons easier. To examine the evolutionary relationship between SYT1, SYT7 and the more distantly related extended-Synaptotagmin (E-SYT) proteins, a phylogenetic tree was generated using the BLOSUM62 matrix and neighbor joining clustering analysis with protein sequences from placozoa (*Trichoplax adhaerens*), invertebrates (*Caenorhabditis elegans*, *Drosophila melanogaster, Ciona intestinalis*) and vertebrates (*Danio rerio, Rattus norvegicus, Homo sapiens, Figure 1B*). Although Trichoplax lacks neurons, it is the earliest metazoan that encodes *Syt* genes and contains both a SYT1 and SYT7 homolog (*Barber et al., 2009*). The phylogenetic tree contains independent clusters that correspond to the SYT1, SYT7 and E-SYT2 protein families. The clustering of SYT1 homologs across evolution correlates with nervous system complexity, with the Trichoplax homolog forming the outlier member of the cluster. Within the SYT7 cluster, *C. elegans* SYT7 is the most distantly related member, with the Trichoplax homolog residing closer within the cluster. *Drosophila* SYT7 is more distant from the vertebrate subfamily clade than is *Drosophila* SYT1 within its subfamily, suggesting SYT7 sequence conservation is not as closely related to nervous system complexity as SYT1. These observations are consistent with SYT7's broader expression pattern and function within neuronal and non-neuronal cells (*MacDougall et al., 2018*).

To compare SYT1 and SYT7 proteins, we performed homology modeling between *Drosophila* SYT7 and the published structure of mammalian SYT7 (*R. norvegicus* SYT7; PBD: 6ANK) (*Voleti et al., 2017*). Key structural features are highly conserved in the homology model, including the eight-stranded β-barrel and the Ca$^{2+}$ binding loops that form the core of C2 domains (*Figure 1C*). In contrast to SYT1, both *Drosophila* and mammalian SYT7 lack the C2B HB helix previously found to have an inhibitory role in SV fusion (*Xue et al., 2010*). We next performed

sequence alignment of SYT proteins from *H. sapiens*, *R. norvegicus* and *D. melanogaster* (*Figure 1—figure supplement 1*). *Drosophila* SYT7 is 59% identical to human SYT7. Comparing the SYT1 and SYT7 subfamilies, the N-terminus encoding the transmembrane domain and linker region has the greatest variability and shares only 21% identity. Within the C2 domains, there is 100% conservation of the negatively charged $Ca^{2+}$ binding residues in the C2 loops. A polybasic stretch in the C2B domain that mediates $Ca^{2+}$-independent PI(4,5)P2-lipid interactions is also conserved. These sequence conservations indicate $Ca^{2+}$-dependent and $Ca^{2+}$-independent membrane binding are key properties of both SYT proteins.

Beyond lipid binding, SYT1's interaction with the SNARE complex is essential for its ability to activate SV fusion. Five key C2B residues (S332, R334, E348, Y391, A455) form the primary interaction site that docks SYT1 onto the SNARE complex (*Guan et al., 2017*; *Zhou et al., 2015*). Four of the five primary SNARE binding residues are not conserved in *Drosophila* SYT7 (*Figure 1C*, *Figure 1—figure supplement 1*). In addition, *Drosophila* and mammalian SYT7 contain specific amino acids substitutions at two of these residues that block SNARE binding and abolish SYT1 function in SV fusion (*Guan et al., 2017*), including C285 (corresponding to *Syt1* mutant R334C) and K299 (corresponding to *Syt1* mutant E348K). A secondary SNARE complex-binding interface on SYT1 is mediated by conserved basic residues at the bottom on the C2B β-barrel (R451/R452 in *Drosophila* R388/R389 in rodents) and is also not conserved in the SYT7 subfamily (*Wang et al., 2016*; *Xue et al., 2010*; *Zhou et al., 2015*). As such, SYT7 is unlikely to engage the SNARE complex via the primary or secondary C2B interface, highlighting a key difference in how the proteins regulate membrane trafficking. Beyond SNARE-binding, 20 nonsynonymous amino acid substitutions are conserved only in the SYT1 or SYT7 subfamilies, suggesting additional interactions have likely diverged during evolution from the common ancestral SYT protein. In summary, SYT1 and SYT7 likely regulate membrane trafficking through distinct mechanisms, consistent with chimeric SYT1/SYT7 rescue experiments in mammals (*Xue et al., 2010*).

## Generation of *Drosophila Syt7* mutations

To assay SYT7 function in *Drosophila* the CRISPR-Cas9 system was used to generate null mutations in the *Syt7* locus. Using a guide RNA targeted near the *Syt7* start codon, several missense mutations were obtained. To disrupt the coding frame of *Syt7*, a single base pair cytosine deletion mutant (*Syt7^M1^*) located seven amino acids downstream of the start codon was used for most of the analysis, with an unaffected Cas9 injection line as control (*Figure 1D*). A Minos transposon insertion in the second coding exon of *Syt7* was also identified from the BDGP gene disruption project (*Bellen et al., 2004*) that generates a premature stop codon before the C2A domain, providing a second independent allele (*Syt7^M2^*) in a distinct genetic background (*Figure 1D*). To characterize the effects of SYT7 overexpression, a UAS-*Syt7* transgene was crossed with the neuronal *elav^C155^*-GAL4 driver. Western blot analysis of adult brain extracts with anti-SYT7 antisera confirmed the absence of SYT7 protein in *Syt7^M1^* and *Syt7^M2^* mutants and a 2.5-fold increase in SYT7 protein levels in *elav^C155^*-GAL4; UAS-*Syt7* (*Figure 1E*). Similar to the loss of SYT7 in mice (*Maximov et al., 2008*), *Drosophila Syt7* null mutants are viable and fertile with no obvious behavioral defects.

## Dose-dependent regulation of neurotransmitter release by SYT7

To assay SYT7's role in synaptic transmission, two-electrode voltage clamp (TEVC) recordings were performed at glutamatergic NMJs from 3rd instar larval motor neurons at segment A3 muscle 6 in 2 mM extracellular $Ca^{2+}$. No significant changes in spontaneous release parameters were identified, as miniature excitatory junctional current (mEJC) amplitude, kinetics and frequency were similar between *Syt7^M1^* mutants, *Syt7^M1^* heterozygotes (*Syt7^M1^*/+) and controls (*Figure 2A–D*). In contrast to spontaneous release, evoked SV fusion (excitatory evoked junctional current (eEJC)) was dramatically enhanced in *Syt7^M1^* single mutants and elevated to an intermediate level in *Syt7^M1^* heterozygotes (*Figure 2E,F*; control: 158.33 ± 19.13 nA, n = 9; *Syt7^M1^*/+: p<0.05, 233.08 ± 19.16 nA, n = 14; *Syt7^M1^*: p<0.005, 262.96 ± 13.01 nA, n = 10). Although evoked release was increased ~2 fold, there was no change in eEJC kinetics in *Syt7^M1^* or *Syt7^M1^*/+ (*Figure 2G,H*). In addition, eEJC half-width was unaffected (*Figure 2I*). Loss of SYT7 increased evoked release regardless of whether quantal content was estimated using eEJC amplitude (which primarily measures synchronous release, 98% increase, *Figure 2J*) or eEJC charge (which measures both synchronous and asynchronous release,

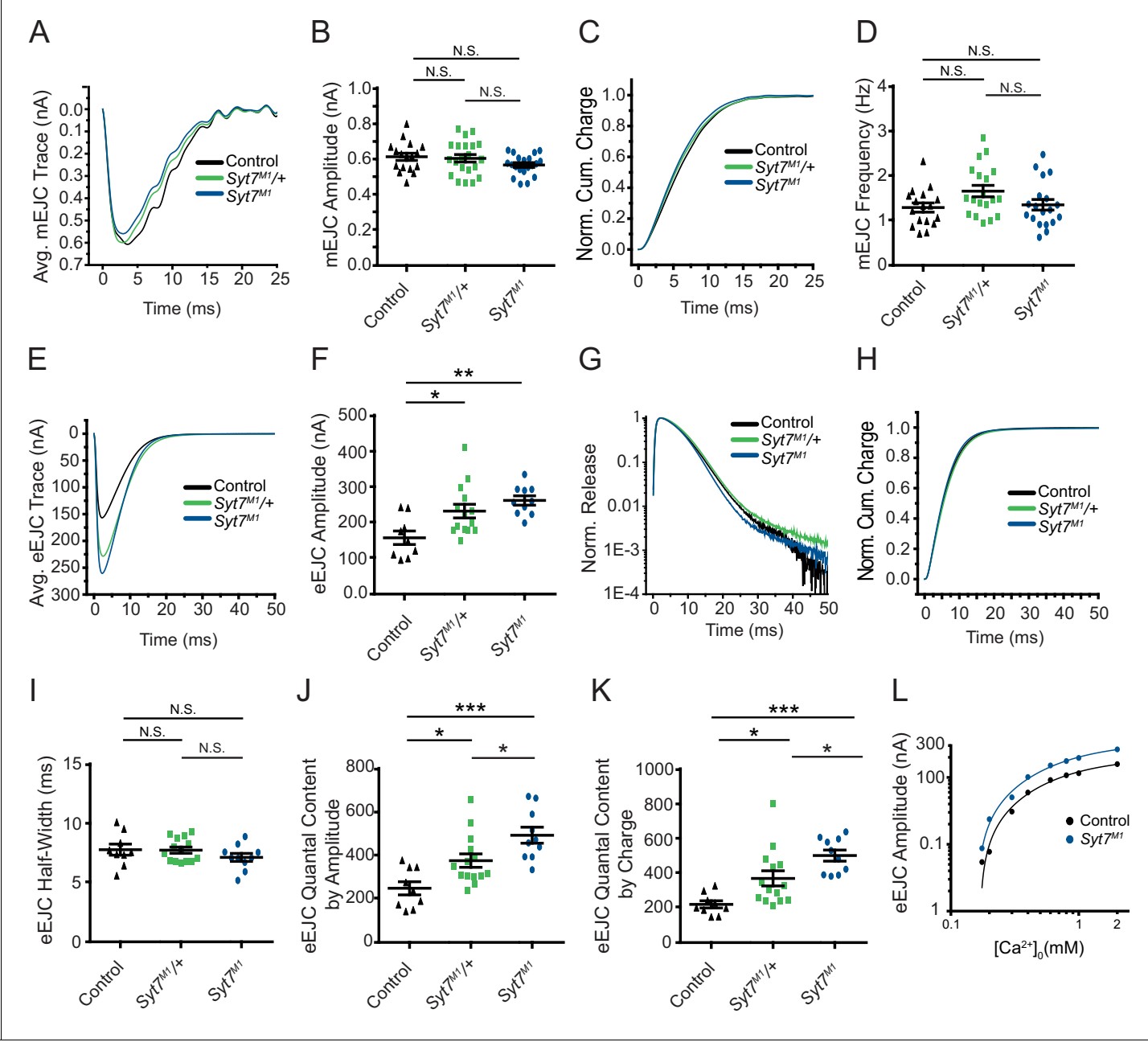

**Figure 2.** *Syt7* mutants and *Syt7/+* heterozygotes display enhanced neurotransmitter release. (**A**) Average mEJC traces in control (black), *Syt7^M1/+* (green) and *Syt7^M1* mutants (blue). (**B**) Quantification of mean mEJC amplitude for the indicated genotypes (control: 0.62 ± 0.020 nA, n = 17; *Syt7^M1/+*: 0.61 ± 0.021 nA, n = 21; *Syt7^M1*: 0.57 ± 0.013 nA, n = 20). (**C**) Normalized cumulative mEJC charge for each genotype. (**D**) Quantification of mean mEJC frequency for the indicated genotypes (control: 1.30 ± 0.10 Hz, n = 17; *Syt7^M1/+*: 1.66 ± 0.13 Hz, n = 19; *Syt7^M1*: 1.36 ± 0.12 Hz, n = 19). (**E**) Average eEJC traces in control (black), *Syt7^M1/+* (green) and *Syt7^M1* (blue). (**F**) Quantification of mean eEJC amplitude for the indicated genotypes. (**G**) Average normalized responses for each genotype plotted on a semi-logarithmic graph to display release components. (**H**) Cumulative release normalized to the maximum response in 2 mM Ca$^{2+}$ for each genotype. (**I**) Quantification of mean eEJC half-width in the indicated genotypes (control: 7.81 ± 0.47 ms, n = 9; *Syt7^M1/+*: 7.77 ± 0.26 ms, n = 14; *Syt7^M1*: 7.15 ± 0.34 ms, n = 10). (**J**) Quantification of evoked quantal content with mEJC amplitude for the indicated genotypes (control: 250.1 ± 30.58 SVs, n = 9; *Syt7^M1/+*: 377.9 ± 31.13, n = 14; *Syt7^M1*: 495.3 ± 36.75, n = 10). (**K**) Quantification of evoked quantal content with mEJC charge for the indicated genotypes (control: 221.3 ± 20.54 SVs, n = 9; *Syt7^M1/+*: 371.6 ± 43.56, n = 14; *Syt7^M1*: 503.6 ± 31.99, n = 10). (**L**) Log-log plot for eEJC amplitudes recorded in 0.175, 0.2, 0.3, 0.4, 0.6, 0.8, 1, and 2 mM extracellular [Ca$^{2+}$] from control (black) and *Syt7^M1* mutants (blue), with a Hill fit for each genotype noted. Recordings were performed from 3$^{rd}$ instar segment A3 muscle 6. Extracellular [Ca$^{2+}$] in **E–K** was 2 mM. Statistical significance was determined using one-way ANOVA (nonparametric) with post hoc Tukey's multiple comparisons test. N.S. = no significant change. Error bars represent SEM.

128% increase, *Figure 2K*). The enhanced evoked release in *Syt7[M1]* was observed over a large range of extracellular [$Ca^{2+}$] spanning from 0.175 to 2 mM (*Figure 2L*). Although the $Ca^{2+}$ response curve shifted leftward over the entire range in *Syt7[M1]*, regression analysis revealed no significant difference in the $Ca^{2+}$ cooperativity of release (control: 2.98 ± 0.17 (n = 7 larvae); *Syt7[M1]*: 2.69 ± 0.50 (n = 7 larvae), p=0.53). We conclude that loss of SYT7 enhances evoked SV release with no major effect on release kinetics or $Ca^{2+}$ cooperativity at *Drosophila* NMJs.

The synaptic levels of SYT7 are likely to be rate-limiting for its ability to regulate synaptic transmission since *Syt7[M1]*/+ heterozygotes displayed an intermediate increase in evoked release compared to *Syt7[M1]* null mutants. To determine if the effects of SYT7 are dosage-sensitive, SYT7 was overexpressed 2.5-fold by driving a UAS-*Syt7* transgene with neuronal *elav[C155]*-GAL4 (*Figure 1E*). Overexpression of SYT7 had no significant effect on spontaneous mEJC kinetics or amplitude (*Figure 3A,B*), similar to the lack of effect in *Syt7* null mutants. However, SYT7 overexpression resulted in a ∼ 2 fold decrease in mEJC frequency (*Figure 2C*, p<0.05), suggesting elevated levels of SYT7 can reduce spontaneous fusion. Unlike the increased evoked release in *Syt7[M1]* and *Syt7[M1]*/+ mutants, SYT7 overexpression caused a striking reduction in eEJC amplitude (*Figure 3D,E*) and eEJC charge (*Figure 3F*), with only mild effects on SV release kinetics (*Figure 3G*). To determine if

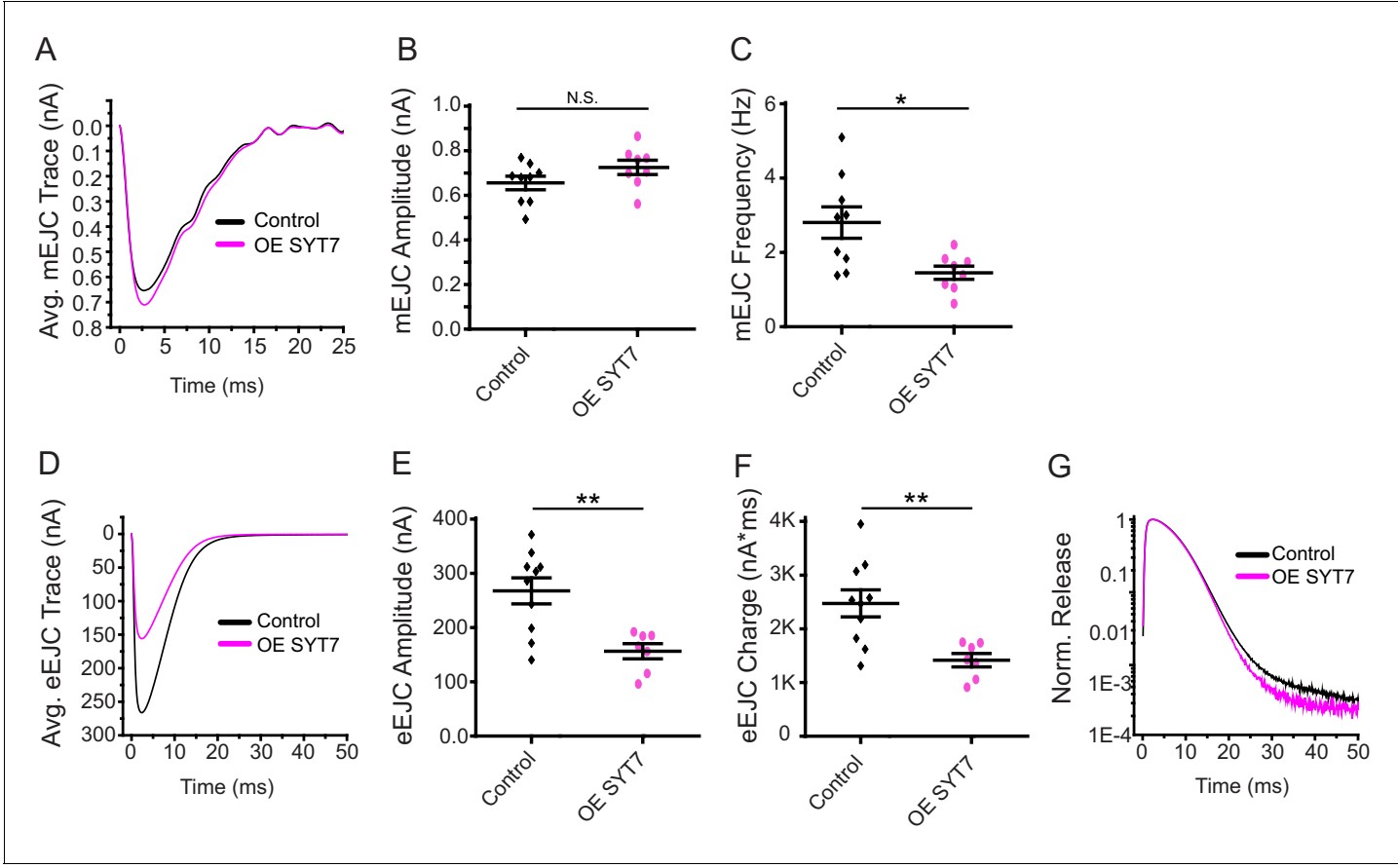

**Figure 3.** Neuronal overexpression of SYT7 reduces spontaneous and evoked SV release. (**A**) Average mEJC traces in control (black) and *elav[C155]*-GAL4; UAS-*Syt7* (OE SYT7, magenta). (**B**) Quantification of mean mEJC amplitudes in the indicated genotypes (control: 0.66 ± 0.03 nA, n = 9; OE SYT7: 0.73 ± 0.03 nA, n = 8). (**C**) Quantification of mean mEJC frequency in the indicated genotypes (control: 2.81 ± 0.42 Hz, n = 9; OE SYT7: 1.45 ± 0.18 Hz, n = 8). (**D**) Average eEJC traces in control (black) and *elav[C155]*-GAL4; UAS-*Syt7* (OE SYT7, magenta). (**E**) Quantification of mean eEJC amplitudes in the indicated genotypes (control: 256.24 ± 22.38 nA, n = 10; OE SYT7: 166.66 ± 10.74 nA, n = 7). (**F**) Quantification of mean eEJC charge in the indicated genotypes (control: $2.5 \times 10^3 ± 0.25 \times 10^3$ nA*ms, n = 10; OE SYT7: $1.4 \times 10^3 ± 0.12 \times 10^3$ nA*ms, n = 7). (**G**) Average normalized responses for each genotype plotted on a semi-logarithmic graph to display release components. Recordings were performed from 3rd instar segment A3 muscle 6 in 2 mM $Ca^{2+}$. Statistical significance was determined with a Mann-Whitney unpaired t-test.

The online version of this article includes the following figure supplement(s) for figure 3:

**Figure supplement 1.** Overexpression of SYT7 in postsynaptic muscles does not disrupt synaptic transmission.

the inhibitory action of SYT7 on SV release is secondary to a presynaptic role, SYT7 was overexpressed postsynaptically using the muscle specific *Mhc*-GAL4 driver. Overexpression of SYT7 in muscles had no effect on eEJC amplitude or kinetics (*Figure 3—figure supplement 1A, B*). We conclude that increased presynaptic SYT7 levels reduce both spontaneous and evoked SV release, indicating SYT7 functions as a negative regulator of neurotransmission.

## Analysis of synaptic structure, AZ morphology and presynaptic $Ca^{2+}$ influx in *Syt7* mutants

To determine if enhanced SV release in the absence of SYT7 results from an increase in AZ number or SV docking, synaptic morphology and ultrastructure at the NMJ was analyzed in $Syt7^{M1}$ mutants. Motor neurons form en passant synaptic boutons along the axon that contain hundreds of individual AZs marked by a central filamentous T-bar composed of the ELKS/CAST homolog Bruchpilot (BRP) (*Ehmann et al., 2014*; *Wagh et al., 2006*). Immunostaining for BRP, the SV-associated protein Complexin (CPX) and a general marker for neuronal membranes (anti-HRP) was performed at muscle 6/7 and muscle 4, the two NMJs analyzed in this study (*Figure 4A–H*). There was no change in the total number of synaptic boutons (*Figure 4C,F*), AZ number defined by BRP puncta (*Figure 4D,G*), or AZ number per muscle surface area (*Figure 4E,H*). To examine the AZ T-bar where SVs cluster, high-resolution structured illumination microscopy (SIM) was performed on larval muscle 4 NMJs following anti-BRP immunostaining. $Syt7^{M1}$ mutants displayed the normal BRP ring architecture and showed no major difference in morphology compared to controls (*Figure 4I*). Individual T-bar size and intraterminal T-bar spacing was quantified in controls and $Syt7^{M1}$ mutants on a Zeiss Airyscan confocal. Although BRP ring structure was intact, $Syt7^{M1}$ mutants displayed a 25% decrease in the average volume of individual BRP-labeled T-bars (*Figure 4J*), but no change in the spacing of T-bars relative to each other (*Figure 4K*). We conclude that loss of SYT7 does not disrupt overall AZ morphology or AZ number, though $Syt7^{M1}$ mutants display a mild decrease in T-bar volume.

To assay if increased release in $Syt7^{M1}$ mutants is secondary to elevated presynaptic $Ca^{2+}$ influx, $Ca^{2+}$ dynamics at NMJs were analyzed using Fluo-4 AM at 3rd instar larval Ib motor terminals at segment A3 muscle 6/7 in control and $Syt7^{M1}$. A stimulation paradigm consisting of three epochs of 10 Hz stimulation for 5 s separated by a 5 s rest period was performed (*Figure 4L*). Maximum presynaptic Flou-4 AM fluorescence during the stimulation paradigm was significantly greater in control than in $Syt7^{M1}$ (control: $10.7 \times 10^6 \pm 1.25 \times 10^6$, n = 11 NMJs from eight larvae; $Syt7^{M1}$: $6.52 \times 10^6 \pm 0.75 \times 10^6$, n = 9 NMJs from eight larvae, p<0.01, *Figure 4L,M*). These data indicate SYT7 does not suppress release by acting as a $Ca^{2+}$ buffer or a negative regulator of $Ca^{2+}$ channel function. Although the mechanism by which presynaptic $Ca^{2+}$ influx is reduced in *Syt7* mutants is unknown, these data are consistent with the reduced AZ BRP volume (*Figure 4J*) and may represent a homeostatic response secondary to the enhanced release in *Syt7* mutants.

To determine if enhanced SV docking could increase the number of SVs available for release in *Syt7* mutants, SV distribution was quantified at larval muscle 6/7 NMJs in control and $Syt7^{M1}$ using transmission electron microscopy (TEM, *Figure 5A*). No change in overall SV density was observed within $Syt7^{M1}$ boutons, indicating SV recycling is largely unperturbed (*Figure 5B*). In contrast to the mild decrease in T-bar area (*Figure 4J*), there was no change in the length of individual AZs defined by the electron dense synaptic cleft (*Figure 5C*, p=0.93; control: 404 ± 34.5 nm, n = 21 AZs from five larvae; $Syt7^{M1}$: 409 ± 28.9 nm, n = 29 AZs from five larvae). To examine docking, SVs in contact with the plasma membrane under the T-bar (within 100 nm, *Figure 5D*) or just outside the T-bar (100 to 400 nm, *Figure 5E*) were quantified. No significant change in the number of SVs docked at the AZ plasma membrane was detected (*Figure 5D–F*), indicating morphological docking defined by EM is not altered in $Syt7^{M1}$ mutants. To quantify SV distribution in the cytoplasm around AZs, SV number was binned into four concentric hemi-circles from 100 to 400 nm radius centered on the T-bar. No significant difference in SV distribution was observed in any bin (*Figure 5G,H*), indicating the morphological distribution of SVs around T-bars is intact in the absence of SYT7. We conclude the enhanced release in $Syt7^{M1}$ mutants is not due to increased AZ number or docked SVs.

## Optical quantal mapping in *Syt7* mutants

Given quantal size (*Figure 2B*), AZ number (*Figure 4D,G*) and SV docking (*Figure 5H*) are unchanged in *Syt7* mutants, increased release probability ($P_r$) at individual AZs is a candidate

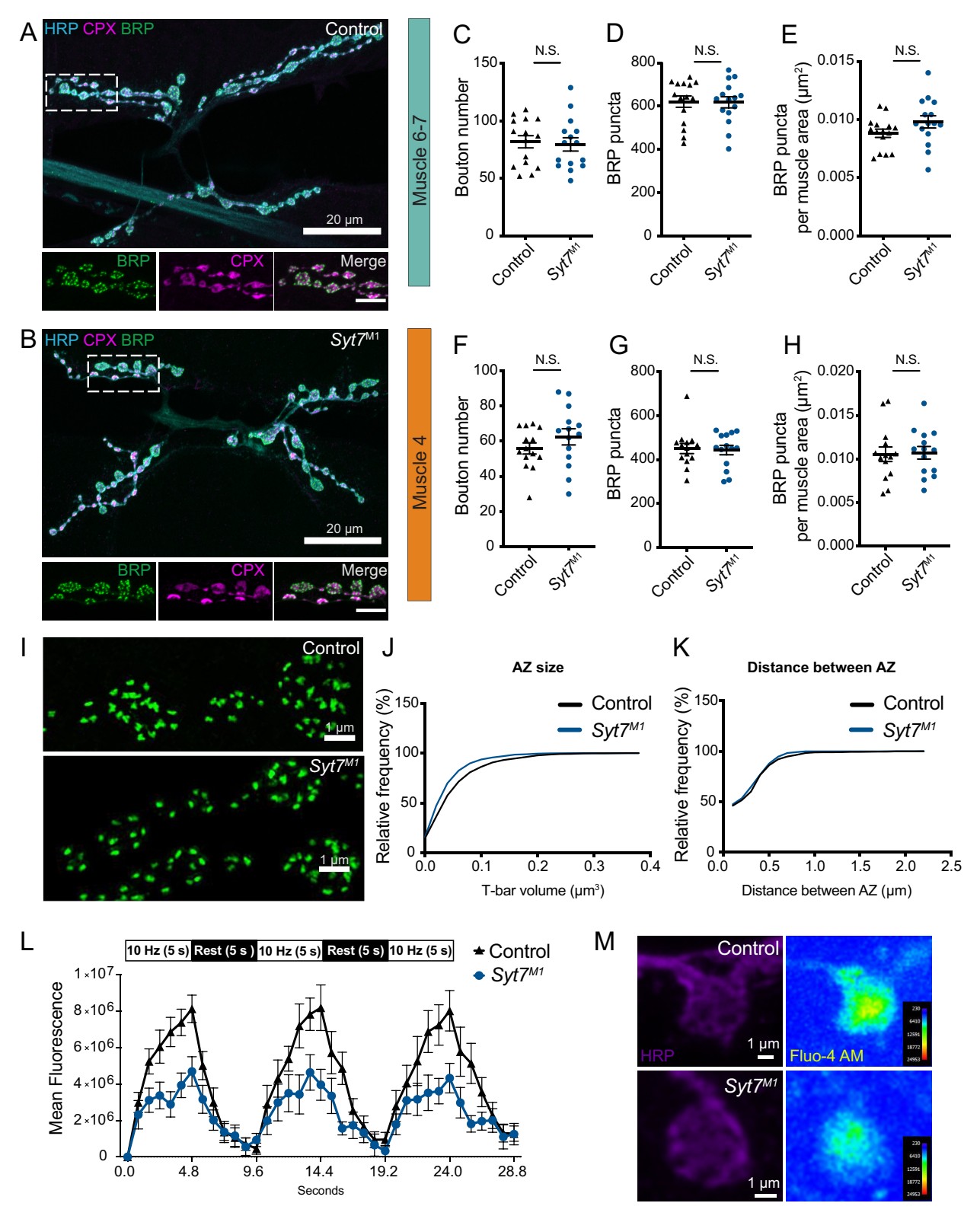

**Figure 4.** Analysis of synaptic morphology in *Syt7* mutants. (**A, B**) Immunocytochemistry of 3<sup>rd</sup> instar muscle 6/7 NMJs with anti-HRP (blue), anti-CPX (magenta) and anti-BRP (green) in control and *Syt7<sup>M1</sup>*. The boxed region is magnified below with channels showing BRP, CPX and the merge. Scale bar = 20 μm for large panels and 2 μm for boxed regions. Synaptic morphology was quantified for 3<sup>rd</sup> instar muscle 6/7 (**C–E**) and muscle 4 (**F–H**) in controls and *Syt7<sup>M1</sup>* mutants. No significant differences were detected in synaptic bouton number (**C, F**); muscle 6/7: p=0.78; control: 81.87 ± 5.301,

*Figure 4 continued on next page*

Figure 4 continued

n = 15; $Syt7^{M1}$: 79.60 ± 5.824, n = 15; muscle 4: p=0.24; control: 55.86 ± 3.141, n = 14; $Syt7^{M1}$: 62.50 ± 4.575, n = 14), BRP puncta (**D, G**), muscle 6/7: p=0.94; control: 621.1 ± 26.28, n = 15; $Syt7^{M1}$: 618.1 ± 25.73, n = 15; muscle 4: p=0.83; control: 450.5 ± 23.25, n = 14; $Syt7^{M1}$: 443.5 ± 21.47, n = 14) or BRP puncta per muscle surface area (**E, H**), muscle 6/7: p=0.13; control: 0.0088 ± 0.0004, n = 15; $Syt7^{M1}$: 0.0098 ± 0.0005, n = 15; muscle 4: p=0.88; control: 0.0105 ± 0.0008, n = 14; $Syt7^{M1}$: 0.0107 ± 0.0007, n = 14). (**I**) Anti-BRP staining at 3rd instar muscle four in control and $Syt7^{M1}$ imaged with SIM microscopy. Scale bar = 1 μm. (**J**) Relative cumulative frequency of AZ T-bar volume defined with anti-BRP staining at 3rd instar muscle 6/7 NMJs (p=0.026; control: 0.055 ± 0.004 $μm^2$, n = 19 NMJs from five larvae; $Syt7^{M1}$: 0.044 ± 0.003 $μm^2$, n = 15 NMJs from four larvae). (**K**) Relative cumulative frequency of T-bar spacing defined by distance between nearest BRP puncta at 3rd instar muscle 6/7 NMJs (p=0.48; control: 0.28 ± 0.016 μm, n = 20 NMJs from five larvae; $Syt7^{M1}$: 0.27 ± 0.014 μm, n = 15 NMJs from four larvae). Statistical significance was determined with Student's t-test. (**L**) Mean fluorescence intensity of Fluo-4 AM in control (black) and $Syt7^{M1}$ mutants (blue) during the indicated stimulation protocol. (**M**) Representative images of synaptic boutons stained with anti-HRP (left), with Fluo-4 AM maximum fluorescence intensity during stimulation shown on the right for control (above) and $Syt7^{M1}$ (below). Scale bar = 1 μm.

mechanism to mediate the elevated quantal content during single stimuli. We previously developed a quantal imaging approach to map AZ $P_r$ at *Drosophila* NMJs by expressing myristoylated GCaMP6s in muscles (*Akbergenova et al., 2018*; *Melom et al., 2013*). Using this approach, $P_r$ maps for evoked release were generated for all AZs from Ib boutons at muscle 4 NMJs in control and $Syt7^{M1}$ mutants (**Figure 6A**). Similar to controls, AZs formed by single motor neurons in $Syt7^{M1}$ displayed heterogeneous $P_r$ (**Figure 6B**). However, $P_r$ distribution was strikingly different between the genotypes, with a greater number of high $P_r$ and fewer low $P_r$ AZs at $Syt7^{M1}$ NMJs (**Figure 6C**). $Syt7^{M1}$ NMJs also had fewer silent AZs that showed no release (control: 19.9%; $Syt7^{M1}$: 4.6%). Overall, mean $P_r$ was increased 2-fold (**Figure 6D**, p<0.01; control: 0.063 ± 0.002, n = 1158 AZs; $Syt7^{M1}$: 0.12 ± 0.004, n = 768 AZs). In contrast, the maximum AZ $P_r$ in the two genotypes was unchanged (**Figure 6D**, control: 0.61; $Syt7^{M1}$: 0.63), indicating an upper limit on release strength for single AZs that is similar between controls and $Syt7^{M1}$. We conclude that the enhanced release in the absence of SYT7 results from an increase in average $P_r$ across the AZ population.

## Loss of SYT7 enhances SV release in *Syt1* null mutants

*Drosophila Syt1* null mutants have dramatically reduced synchronous SV fusion and enhanced asynchronous and spontaneous release (*Jorquera et al., 2012*; *Lee et al., 2013*; *Yoshihara et al., 2010*; *Yoshihara and Littleton, 2002*). We generated *Syt1; Syt7* double mutants to determine if SYT7 mediates the residual asynchronous release present in *Syt1* nulls. A complete loss of asynchronous release in *Syt1; Syt7* double mutants should occur if SYT7 functions as the sole asynchronous $Ca^{2+}$ sensor, while a reduction in release is expected if it is one of several sensors mediating the residual synaptic transmission in *Syt1*. Animals lacking SYT1 were obtained by crossing an intragenic *Syt1* deletion ($Syt1^{N13}$) with a point mutant containing an early stop codon ($Syt1^{AD4}$), an allelic combination referred to as $Syt1^{Null}$. Loss of SYT1 results in lethality throughout development, although some $Syt1^{Null}$ mutants survive to adulthood when cultured under special conditions (*Loewen et al., 2001*). Surviving $Syt1^{Null}$ adults are severely uncoordinated and die within several days. Quantification of survival rates demonstrated 45.3% of $Syt1^{Null}$ mutants survived from the 1st instar to the pupal stage, with 44.1% of mutant pupae surviving to adulthood (n = 5 groups with >40 starting animals each). In contrast, 5.6% of $Syt1^{Null}$; $Syt7^{M2}$ double mutants (referred to as $Double^{Null}$) survived from the 1st instar to the pupal stage, and 6.6% of mutant pupae survived to adulthood (n = 6 groups with >80 animals each). Western blot analysis confirmed loss of both proteins in $Double^{Null}$ mutants and demonstrated no change in expression of SYT1 or SYT7 in the absence of the other family member in individual null mutant backgrounds (**Figure 7A**). Although loss of both SYTs caused synergistic defects in survival, residual synaptic transmission must exist given some $Double^{Null}$ mutants survive.

To assay synaptic transmission, recordings were performed from 3rd instar larval muscle 6 in 2 mM extracellular $Ca^{2+}$ in $Syt1^{Null}$ and $Double^{Null}$ mutants. No change in spontaneous mEJC amplitude or kinetics was found between the two genotypes (**Figure 7B**), indicating postsynaptic sensitivity, neurotransmitter loading, and fusion pore dynamics were not disrupted by loss of SYT7. However, a ~ 2 fold increase in mEJC frequency was observed in the $Double^{Null}$ compared to $Syt1^{Null}$ (**Figure 7C**, p<0.001; $Syt1^{null}$: 2.99 ± 0.23 Hz, n = 16; $Double^{Null}$: 5.33 ± 0.42 Hz, n = 14), demonstrating loss of both SYTs enhances the already elevated spontaneous release rate found in $Syt1^{Null}$ mutants alone. Measurements of evoked release revealed both amplitude and charge transfer were

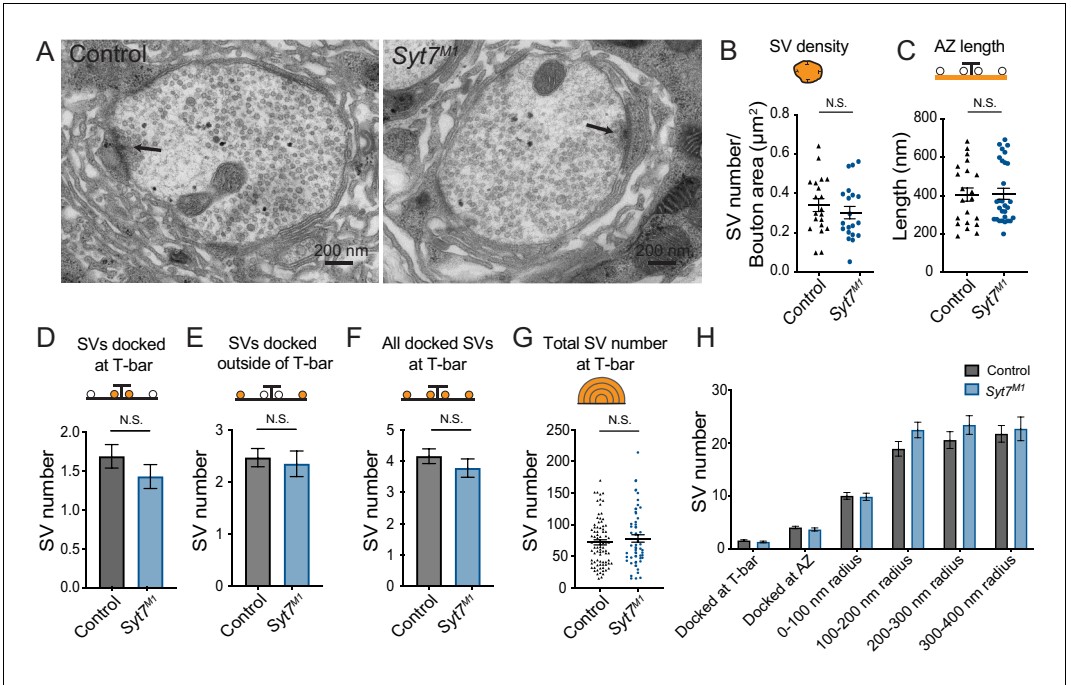

**Figure 5.** Ultrastructural analysis of SV distribution in *Syt7* mutants. (**A**) Representative EM micrographs of muscle 6/7 synaptic boutons in control and *Syt7^M1* 3^rd instar larvae. An AZ with its associated electron dense T-bar is denoted with an arrowhead in each micrograph. (**B**) Quantification of SV density (p=0.41; control = 0.34 ± 0.033 SVs/μm², n = 20; *Syt7^M1* = 0.30 ± 0.031 SVs/μm², n = 20). (**C**) Quantification of AZ length defined by the electron dense synaptic cleft (p=0.93; control: 404 ± 34.5 nm, n = 21 AZs from five larvae; *Syt7^M1*: 409 ± 28.9 nm, n = 29 AZs from five larvae). (**D**) Quantification of SVs docked within 100 nm of the T-bar (p=0.41; control = 1.69 ± 0.15 SVs n = 84; *Syt7^M1* = 1.43 ± 0.15 SVs, n = 58). (**E**) Quantification of SVs docked within 100–400 nm of the T-bar (p=0.68; control = 2.46 ± 0.17 SVs n = 84; *Syt7^M1* = 2.35 ± 0.25 SVs, n = 58). (**F**) Quantification of all docked SVs at 0–400 nm from the T-bar (p=0.31; control = 4.16 ± 0.23 SVs n = 84; *Syt7^M1* = 3.78 ± 0.29 SVs, n = 58). (**G**) Quantification of all SVs within a 400 nm radius from the T-bar (p=0.38; control = 71.98 ± 4.05 SVs n = 84; *Syt7^M1* = 78.12 ± 5.89 SVs, n = 58). (**H**) Quantification of SV distribution at AZs in control and *Syt7^M1* mutants. Statistical significance was determined with Student's t-test.

increased ~2 fold in *Double^Null* compared to *Syt1^Null* mutants (**Figure 7D–F**; eEJC amplitude: p<0.001; *Syt1^Null*: 3.18 ± 0.4 nA, n = 15; *Double^Null*: 6.12 ± 0.62 nA, n = 13; eEJC charge: p<0.05; *Syt1^Null*: 33.2 ± 4.4 nA*ms, n = 15; *Double^Null*: 52.6 ± 5.8 nA*ms, n = 13). In addition, more SVs fused in the first 15 ms following stimulation (**Figure 7G,H**), with less SVs available for release later in the response. *Double^Null* mutants also had a reduced rate of evoked failures following nerve stimulation compared to *Syt1^Null*, consistent with an increased probability of SV release (**Figure 7I**, p<0.01; *Syt1^Null*: 21.1 ± 3.5% failure rate, n = 17; *Double^Null*: 7.4 ± 3.4% failure rate, n = 14). These results indicate SYT7 does not mediate the residual release found in the absence of SYT1. We conclude SYT7 negatively regulates SV release with or without SYT1 present at the synapse.

## Short-term facilitation does not require SYT7

Although these results indicate SYT7 is a not a key asynchronous Ca²⁺ sensor in *Drosophila* the protein has also been implicated as the Ca²⁺ sensor for facilitation (*Chen et al., 2017*; *Jackman et al., 2016*; *Turecek and Regehr, 2018*), a short-term form of presynaptic plasticity that results in enhanced SV fusion during closely-spaced stimuli. To examine facilitation, [Ca²⁺] was lowered from 2 mM to 0.175 mM or 0.2 mM to identify conditions where the initial $P_r$ was matched between control and *Syt7* mutants. In 0.175 mM Ca²⁺, controls displayed an 11% failure ratio in response to single action potentials, while *Syt7^M1* had no failures (**Figure 8A**). In 0.2 mM Ca²⁺, neither genotype had failures (**Figure 8A**), although evoked release was increased 3-fold in *Syt7^M1* (**Figure 7B,C**, p<0.01, control: 7.73 ± 1.5 nA, n = 9; *Syt7^M1*: 23.72 ± 6.2 nA, n = 9). In contrast, EJC amplitude was not

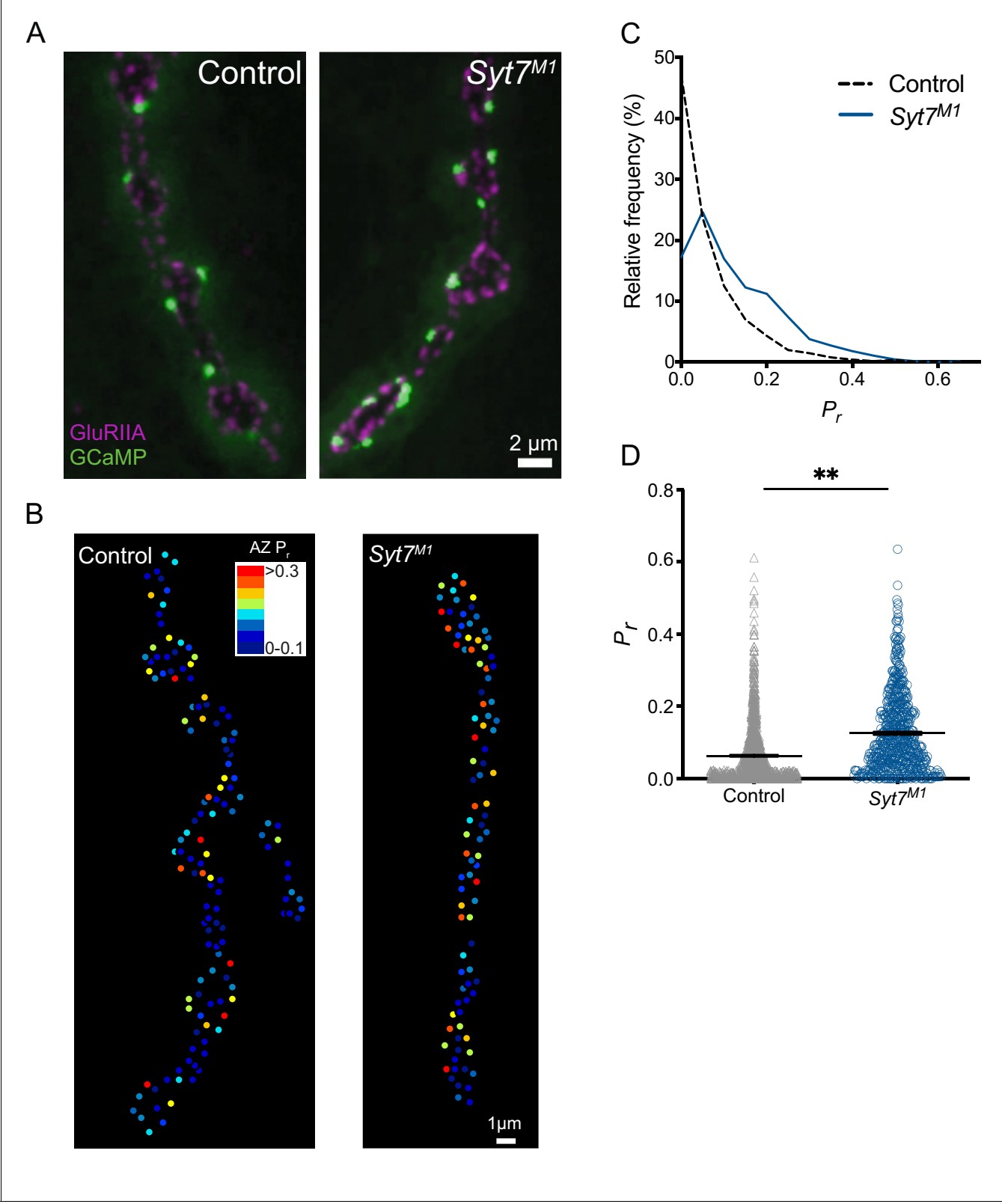

**Figure 6.** Quantal imaging reveals elevated release probability across the AZ population in *Syt7* mutants. (**A**) Representative images of GluRIIA positive PSDs (red) and postsynaptic myr-GCaMP6 flashes (green) in response to evoked stimulation in control and *Syt7M1* mutants. (**B**) $P_r$ heatmaps for muscle 4 NMJs generated following 0.3 Hz stimulation for 5 min in control and *Syt7M1* mutants. The $P_r$ color map is displayed in the upper right. (**C**) Frequency

*Figure 6 continued on next page*

*Figure 6 continued*

distribution of AZ $P_r$ after a 0.3 Hz 5 min stimulation for control (black dashed line) and *Syt7^{M1}* (blue line). (**D**) Quantification of mean AZ $P_r$ for the two genotypes (p≤0.01, Student's t-test; control: 0.063 ± 0.002, n = 1158; *Syt7^{M2}*: 0.12 ± 0.004, n = 768).

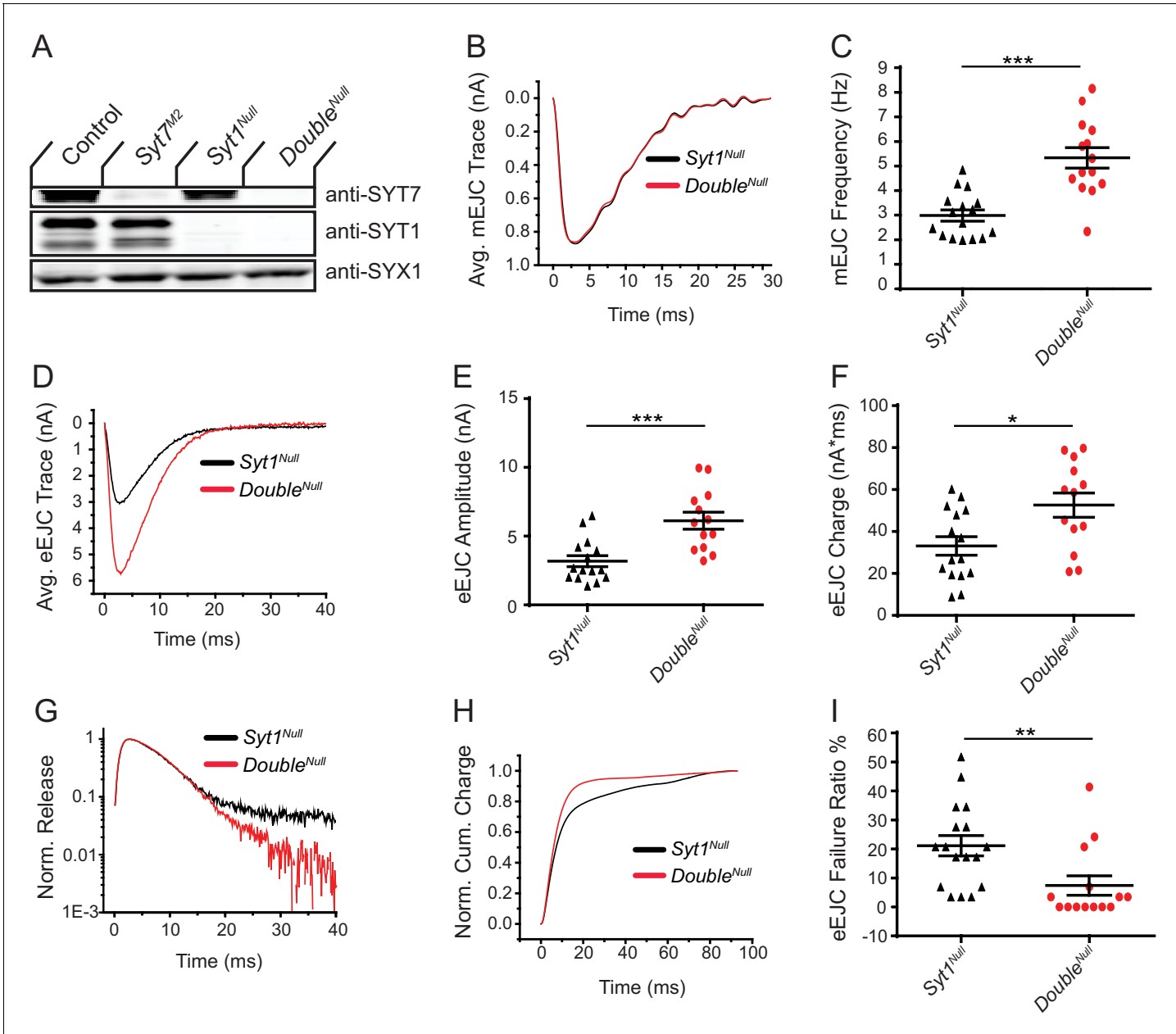

**Figure 7.** Loss of SYT7 enhances the residual release observed in *Syt1* null mutants. (**A**) Western blot of head extracts from control, *Syt7^{M2}*, *Syt1^{Null}* and *Syt1^{null}*; *Syt7^{M2}* (*Double^{Null}*) probed with anti-SYT7, anti-SYT1 and anti-SYX1 (loading control). SYT1 migrates as a doublet at 55 and 70 kD (**Littleton et al., 1993a**). (**B**) Average mEJC traces in *Syt1^{Null}* (black trace) and *Double^{Null}* (red trace) mutants obtained by summing all mEPSC events under the first peak distribution. (**C**) Quantification of mean mEJC frequency for the indicated genotypes. (**D**) Average eEJC traces in *Syt1^{Null}* (black trace) and *Double^{Null}* (red trace). (**E**) Quantification of mean eEJC amplitude for the indicated genotypes. (**F**) Quantification of mean eEJC charge for the indicated genotypes obtained by measuring total release over time. (**G**) Average normalized responses for each genotype plotted on a semi-logarithmic graph to display release components. (**H**) Cumulative release normalized to the maximum response in 2 mM $Ca^{2+}$ for each genotype. Each trace was adjusted to a double exponential fit. (**I**) Quantification of eEJC failure ratio (%) in the indicated genotypes. Recordings were performed from 3rd instar segment A3 muscle 6 in 2 mM extracellular $Ca^{2+}$. Statistical significance was determined with the Mann-Whitney unpaired t-test.

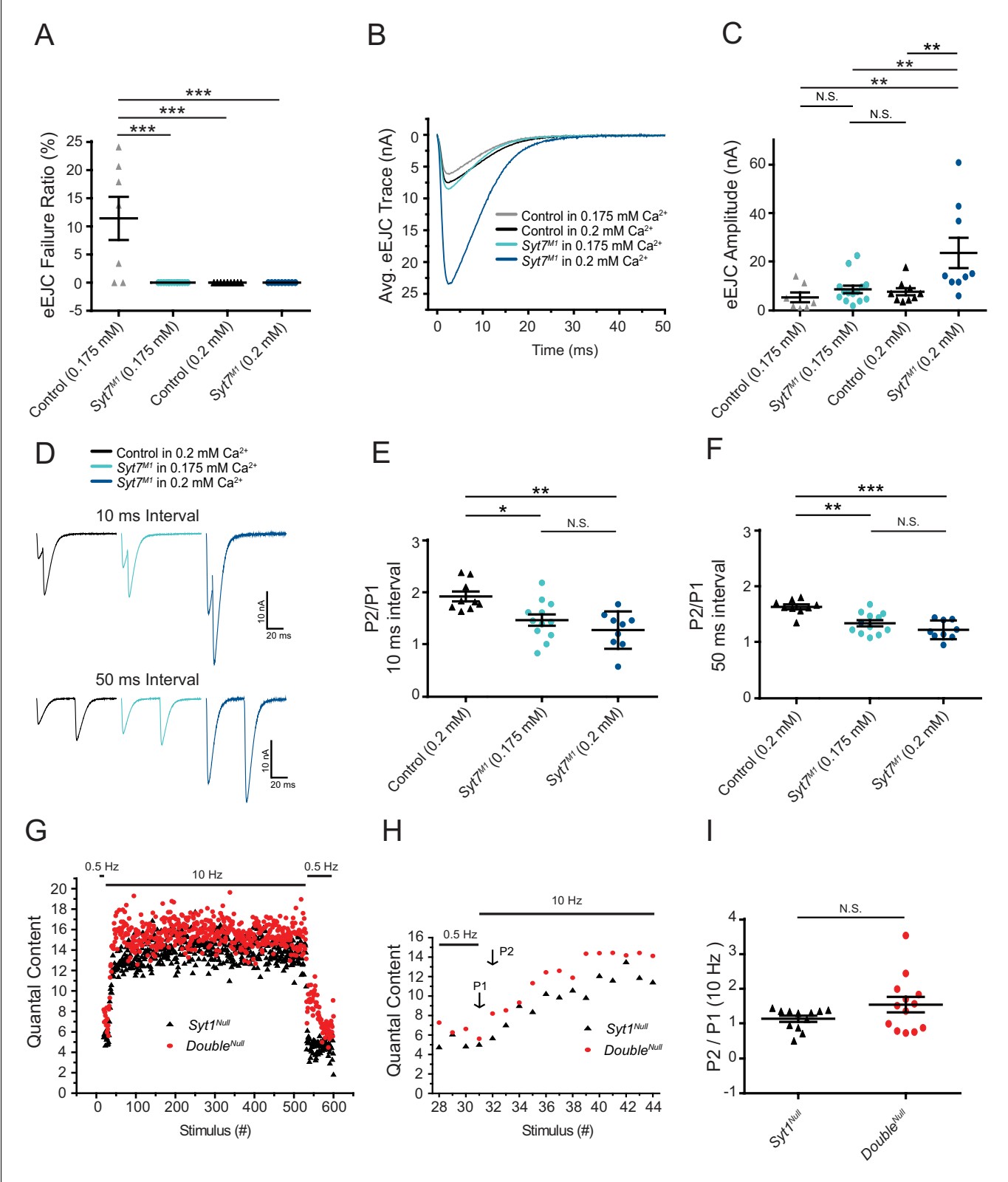

**Figure 8.** Short-term synaptic facilitation can occur without SYT7 or SYT1. (**A**) Quantification of eEJC failure ratio (%) in the indicated genotypes. (**B**) Average eEJC traces recorded in 0.175 mM Ca²⁺ (control, grey; *Syt7^M1*, light blue) or 0.2 mM Ca²⁺ (control, black; *Syt7^M1*, dark blue). (**C**) Quantification of mean eEJC amplitude for the indicated genotypes (0.175 mM Ca²⁺: control, 5.42 ± 2.0 nA, n = 7; *Syt7^M1*, 8.70 ± 1.6 nA, n = 14; 0.2 mM Ca²⁺: control, 7.73 ± 1.5 nA, n = 9; *Syt7^M1*, 23.72 ± 6.2 nA, n = 9). (**D**) Representative eEJC traces to 10 ms or 50 ms paired-pulse stimuli recorded in 0.2 mM Ca²⁺

*Figure 8 continued on next page*

*Figure 8 continued*

(control, black; $Syt7^{M1}$, dark blue) or 0.175 mM $Ca^{2+}$ ($Syt7^{M1}$, light blue). **(E)** Quantification of facilitation (P2/P1) at 10 ms interval for the indicated genotypes (0.2 mM $Ca^{2+}$: control, 1.93 ± 0.095, n = 9; $Syt7^{M1}$, 1.28 ± 0.12, n = 9; 0.175 mM $Ca^{2+}$: $Syt7^{M1}$, 1.47 ± 0.11, n = 12). **(F)** Quantification of facilitation (P2/P1) at 50 ms interval for the indicated genotypes (0.2 mM $Ca^{2+}$: control, 1.64 ± 0.043, n = 9; $Syt7^{M1}$, 1.23 ± 0.056, n = 9; 0.175 mM $Ca^{2+}$: $Syt7^{M1}$, 1.34 ± 0.054, n = 12). Statistical significance was determined using one-way ANOVA (nonparametric) with post hoc Tukey's multiple comparisons test for panels A-F. **(G)** Average eEJC quantal content determined from mEJC charge in 2 mM $Ca^{2+}$ during a 10 Hz stimulation paradigm (30 stimuli at 0.5 Hz, 500 stimuli at 10 Hz, and return to 0.5 Hz) in $Syt1^{Null}$ (black) and $Double^{Null}$ (red). **(H)** Average quantal content for the last four responses of 0.5 Hz stimulation and the first 14 responses during 10 Hz stimulation in $Syt1^{Null}$ (black) and $Double^{Null}$ (red). P1 denotes the 1st response and P2 the 2nd response to 10 Hz stimulation. **(I)** Quantification of P2/P1 ratio in $Syt1^{Null}$ (black, 1.15 ± 0.089, n = 12) and $Double^{Null}$ (red, 1.55 ± 0.22, n = 13) at onset of 10 Hz stimulation. Statistical significance was determined with a Mann-Whitney unpaired t-test for panels H and I.

statistically different between control in 0.2 mM $Ca^{2+}$ (7.73 ± 1.5 nA, n = 9) and $Syt7^{M1}$ in 0.175 mM $Ca^{2+}$ (8.70 ± 1.6 nA, n = 9). Facilitation was assayed in these conditions where initial $P_r$ was comparable. Control and $Syt7^{M1}$ mutants displayed robust facilitation to paired pulses separated by 10 or 50 ms at both $Ca^{2+}$ concentrations (*Figure 8D*). A modest reduction in paired-pulse ratio was observed in $Syt7^{M1}$ at 0.175 $Ca^{2+}$ compared to control at 0.2 mM $Ca^{2+}$ (*Figure 8E,F*, p<0.05; 10 ms interval: 31% decrease; 50 ms interval: 22% decrease). These data indicate SYT7 is not the sole effector of facilitation. The mild defect in $Syt7$ mutants could be due to a partially redundant role for SYT7 in facilitation or secondary to differences in $Ca^{2+}$ available to activate the true facilitation sensor. Given [$Ca^{2+}$] was lowered in $Syt7^{M1}$ to match initial $P_r$ between the genotypes, the latter hypothesis is more likely.

To determine if short-term facilitation could be elicited in the absence of both SYT1 and SYT7, a 10 Hz stimulation train in 2.0 mM $Ca^{2+}$ was given to $Double^{Null}$ mutants and eEJC responses were compared to $Syt1^{Null}$ mutants alone. Similar to the increased quantal content to single action potentials, $Double^{Null}$ mutants displayed larger facilitating responses during the early phase of stimulation (*Figure 8G–I*; cumulative average release for 10 stimuli: $Syt1^{Null}$ (n = 12): 87 ± 7.0 quanta; $Double^{Null}$ (n = 13): 109 ± 9.9 quanta; 20 stimuli: $Syt1^{Null}$: 209 ± 13.8 quanta; $Double^{Null}$: 261 ± 22.6 quanta; 50 stimuli: $Syt1^{Null}$: 594 ± 34.5 quanta; $Double^{Null}$: 745 ± 56.2 quanta, p<0.03). These results indicate short-term facilitation can occur in the absence of both SYT1 and SYT7, and is enhanced during the early phases of stimulation, consistent with SYT7 negatively regulating SV fusion with or without SYT1.

## Syt7 mutants have access to a larger pool of fusogenic SVs but maintain a normal rate of SV endocytosis at steady-state

Enhanced SV release in $Syt7$ mutants could reflect increased fusogenicity of the entire SV population or conversion of a non-fusogenic SV pool into one capable of release in the absence of SYT7. To test whether SYT7 normally renders a pool of SVs non-fusogenic, 1000 stimuli at 10 Hz were applied in 2 mM $Ca^{2+}$ at 3rd instar muscle 6 NMJs to deplete the readily releasable pool (RRP) and drive SV cycling to steady-state. The total number of released SVs and the SV recycling rate was then measured. Both control and $Syt7^{M1}$ eEJCs depressed during the stimulation train. However, SV release in $Syt7^{M1}$ mutants remained elevated over much of the initial stimulation (*Figure 9A*) and the integral of release during the train was greater than controls (*Figure 9B*), indicating $Syt7$ nulls have access to more fusogenic SVs. SV release rate in $Syt7^{M1}$ eventually reached the same level as control following depletion of the RRP (*Figure 9C*, control quantal content: 131.5 ± 10.7, n = 7; $Syt7^{M1}$ quantal content: 123.1 ± 10.5, n = 8). We conclude that SV endocytosis and recycling rate is SYT7-independent at steady-state, although $Syt7^{M1}$ mutants contain a larger RRP available for fusion.

To further examine SV recycling, FM1-43 dye uptake and release assays were performed in control and $Syt7^{M1}$ mutants at 3rd instar muscle 6/7 NMJs. At low stimulation rates (0.5 Hz), $Syt7^{M1}$ mutants took up significantly more FM1-43 dye than controls (*Figure 9D,F*), consistent with the increased SV release observed by physiology. In contrast, no significant difference in FM1-43 uptake was found following high frequency 10 Hz stimulation for 500 stimuli (*Figure 9E,G*). These data suggest previously exocytosed SVs re-enter the RRP more often in the absence of SYT7 given the normal recycling rate (*Figure 9C*). Consistent with this hypothesis, no change in FM1-43 release was detected with high [$K^+$] stimulation following 10 Hz loading (*Figure 9H*). Together with the electrophysiology data, we conclude $Syt7$ mutants have a larger RRP, but no changes in SV endocytosis.

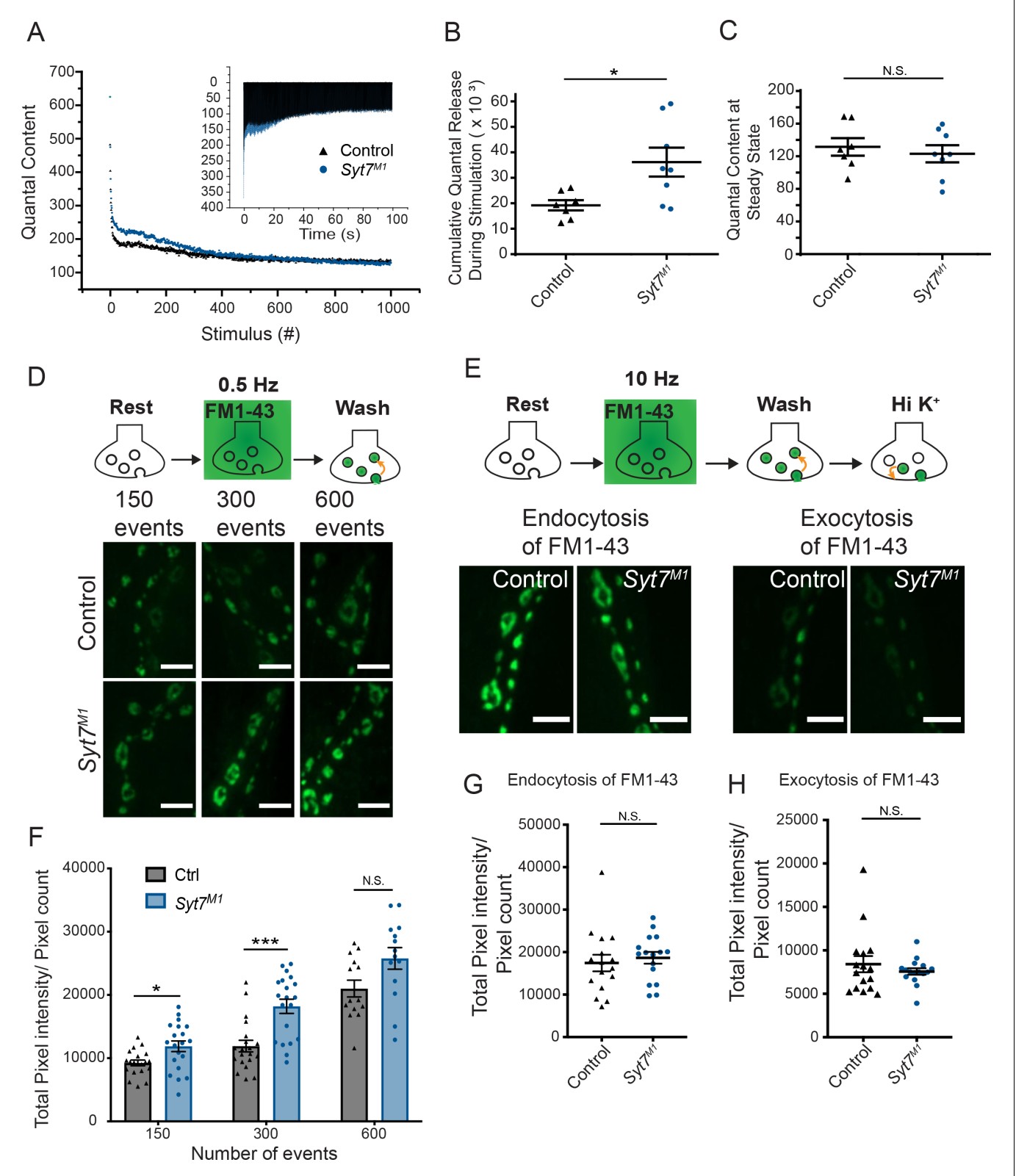

**Figure 9.** *Syt7* mutants have a larger releasable pool of SVs and normal endocytosis. (**A**) Representative mean eEJC quantal content determined by mEJC charge during 1000 stimuli at 10 Hz in 2 mM Ca²⁺ in control (black) and *Syt7*ᴹ¹ (blue). The inset shows representative eEJC traces in control (black) and *Syt7*ᴹ¹ (blue). (**B**) Quantification of average cumulative quanta released during the 1000 stimuli at 10 Hz tetanic stimulation in control (black, 19.21K ± 2.88K, n = 7) and *Syt7*ᴹ¹ (blue, 36.18K ± 5.67K, n = 8). (**C**) Quantification of average quantal content at steady-state release at the end of the

*Figure 9 continued on next page*

*Figure 9 continued*

10 Hz stimulation in control (black, 131.54 ± 10.71, n = 7) and *Syt7^{M1}* (blue, 123.05 ± 10.47, n = 8). Statistical significance for B and C was determined with a Mann-Whitney unpaired t-test. (D) FM1-43 loading in control and *Syt7^{M1}* larvae at muscle 6/7 NMJs in 2 mM $Ca^{2+}$ following 150, 300 or 600 stimuli delivered at 0.5 Hz. (E) FM1-43 loading with 500 stimuli at 10 Hz in 2 mM $Ca^{2+}$ and FM1-43 unloading with high $K^+$ (90 mM) in control and *Syt7^{M1}* larvae at muscle 6/7 NMJs. (F) Quantification of FM1-43 loading following 150, 300 or 600 stimuli delivered at 0.5 Hz. (G) Quantification of FM1-43 loading after 500 stimulati at 10 Hz. (H) Quantification of FM1-43 unloading with high $K^+$ (90 mM). Statistical significance was determined with Student's t-test for F-H. Scale bar = 5 µm.

## *Syt7* mutants have enhanced refilling of the readily-releasable SV pool independent of endocytosis

To probe how SYT7 regulates SV cycling and the transition between distinct SV pools, eEJC recovery kinetics following high frequency stimulation were characterized. A paradigm consisting of 30 stimuli at 0.5 Hz, 500 stimuli at 10 Hz and a final 50 stimuli at 0.5 Hz was given to *Syt7^{M1}* mutants, *Syt7^{M1}*/+ heterozygotes and controls in 2 mM $Ca^{2+}$ (*Figure 10A*). During 0.5 Hz stimulation, *Syt7^{M1}* and *Syt7^{M1}*/+ displayed elevated levels of release. Following the onset of high frequency stimulation, *Syt7^{M1}* and *Syt7^{M1}*/+ synapses depressed while controls displayed a mild facilitation before quickly transitioning to depression (*Figure 10B*). Remarkably, *Syt7^{M1}* and *Syt7^{M1}*/+ displayed an extremely rapid recovery of eEJC amplitude and quantal content during the 2 s interval following termination of the 10 Hz train compared to controls (*Figure 10C*). A similar rapid recovery was observed in *Syt7^{M1}* after 2000 stimuli were given at 10 Hz to fully deplete the RRP and normalize release rates to control levels (*Figure 10—figure supplement 1A–C*). These observations suggest SYT7 also functions to reduce SV entry into the RRP, while negatively regulating release of newly regenerated SVs. The enhanced refilling of the RRP did not require SYT1 function, as *Double^{Null}* mutants also displayed larger eEJCs than *Syt1^{Null}* alone during the recovery from a 10 Hz stimulation train (*Figure 8G*).

The partial elevation in RRP refilling rate at *Syt7^{M1}*/+ synapses indicates the amount of SYT7 in the presynaptic terminal regulates SV entry into the releasable pool. To determine if RRP refilling is dosage-sensitive, the stimulation paradigm above (0.5 Hz/10 Hz/0.5 Hz) was applied to SYT7 overexpression larvae (*elav^{C155}*-GAL4; UAS-*Syt7*) in 2 mM $Ca^{2+}$. Presynaptic overexpression of SYT7 had the opposite effect of *Syt7* mutants and *Syt7*/+ heterozygotes, not only reducing eEJC amplitude at 0.5 Hz, but greatly limiting the ability of SVs to re-enter the RRP following termination of the 10 Hz stimulation train (*Figure 10—figure supplement 2A-C*). We conclude that SYT7 limits release in a dosage-sensitive manner by negatively regulating the number of SVs available for fusion and slowing recovery of the RRP following stimulation.

To determine if increased RRP refilling in *Syt7^{M1}* requires an enhanced rate of SV endocytosis or is mediated through refilling from a pre-existing SV pool, recordings were repeated in the presence of the proton pump inhibitor bafilomycin. Bafilomycin blocks neurotransmitter reloading of newly endocytosed SVs and should eliminate the enhanced refilling of the RRP if recycling is essential. Alternatively, if SVs are recruited more rapidly from pre-existing pools, bafilomycin would not abolish the enhanced recovery. The same 0.5 Hz/10 Hz/0.5 Hz paradigm was applied in three successive epochs in the presence of 4 uM bafilomycin or DMSO (control) in the bath solution. As expected, bafilomycin progressively reduced eEJC amplitude throughout the experiment and eliminated most evoked responses during the 3rd stimulation epoch (*Figure 10D*). *Syt7^{M1}* mutants displayed a similar fold-enhancement in the recovery of the RRP in the presence of bafilomycin, though the absolute numbers of SVs re-entering the pool decreased following the 2nd 10 Hz stimulation as the number of neurotransmitter-containing SVs declined (*Figure 10E,F*). We conclude that the rapid refilling of the RRP can occur from pre-existing SV pools. In addition to reducing fusogenicity of SVs already docked at the AZ, these data indicate SYT7 regulates transition kinetics between vesicle pools by reducing the number of SVs moving from the reserve pool to the RRP.

## SYT7 localizes to an internal membrane network within the peri-AZ that resides in proximity to multiple presynaptic compartments

Defining the subcellular localization of SYT7 could help elucidate how it modulates SV dynamics. SYT7 could be a resident protein of the SV pool it regulates or reside on an alternative compartment that exerts control over a subset of SVs. To examine the subcellular localization of SYT7, an RFP tag

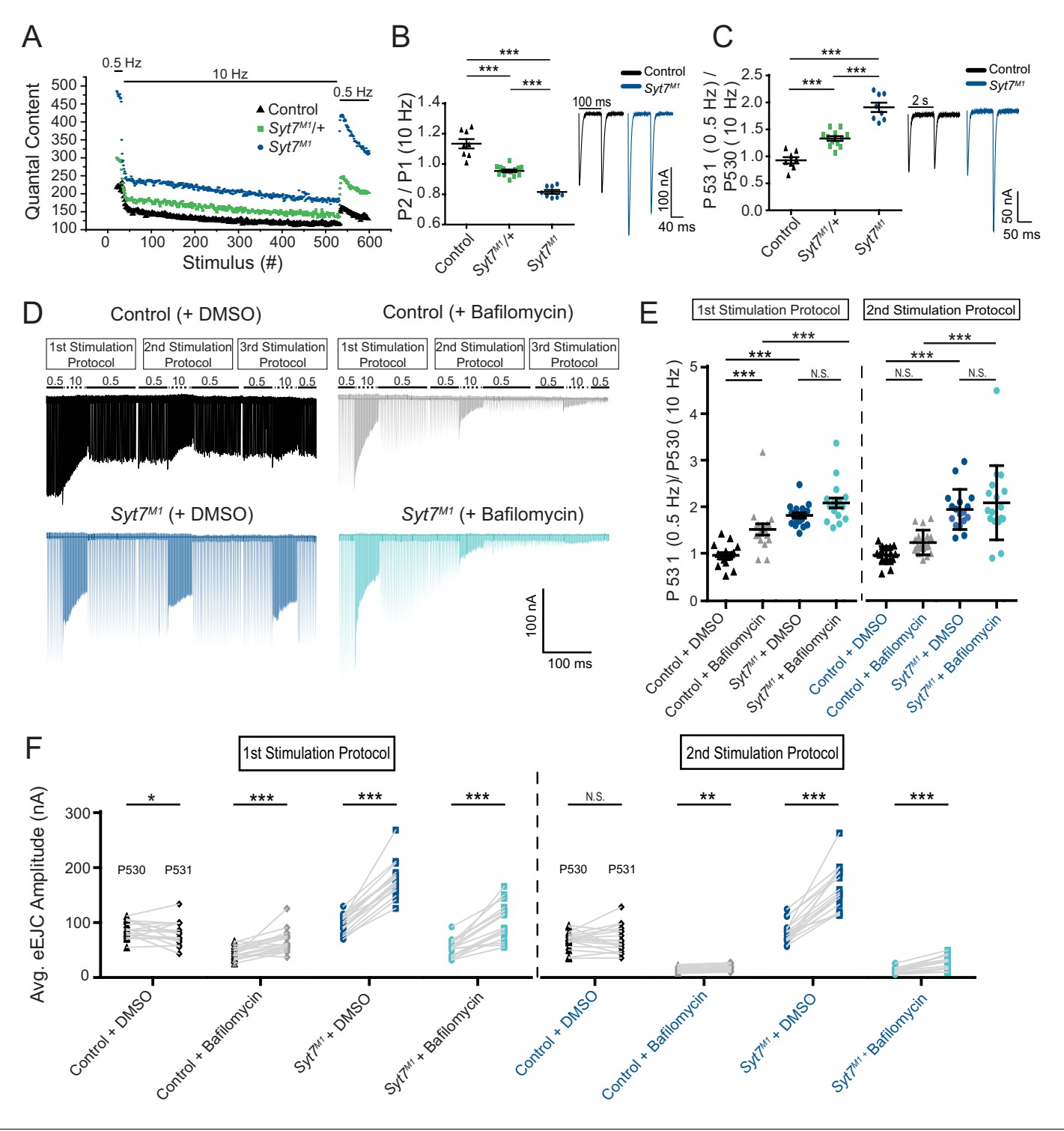

**Figure 10.** *Syt7* mutants have enhanced refilling of the RRP that does not require endocytosis. (**A**) Average eEJC quantal content during the indicated stimulation protocol in 2 mM external Ca$^{2+}$ for control (black), *Syt7*$^{M1/+}$ (green) and *Syt7*$^{M1}$ (blue). (**B**) Quantification of P2/P1 ratio (P1 = 1$^{st}$ response to 10 Hz, P2 = 2$^{nd}$ response to 10 Hz) in control (black, 1.13 ± 0.03, n = 8), *Syt7*$^{M1/+}$ (green, 0.95 ± 0.009, n = 14) and *Syt7*$^{M1}$ (blue, 0.82 ± 0.01, n = 8). Representative eEJC traces of P1 and P2 for control (black) and *Syt7*$^{M1}$ (blue) are shown on the right. (**C**) Quantification of P531/P530 ratio (P530 is the last response to 10 Hz and P531 is the 1$^{st}$ response to 0.5 Hz stimulation delivered 2 s after P530) in control (black, 0.93 ± 0.06, n = 8), *Syt7*$^{M1/+}$ (green, 1.33 ± 0.04, n = 12) and *Syt7*$^{M1}$ (blue, 1.91 ± 0.09, n = 8). Representative eEJC traces of P530 and P531 for control (black) and *Syt7*$^{M1}$ (blue) are shown on the right. (**D**) Representative eEJC traces for control with DMSO (black) or 4 μM bafilomycin (gray) and *Syt7*$^{M1}$ with DMSO (dark blue) or 4 μM

*Figure 10 continued on next page*

*Figure 10 continued*

bafilomycin (light blue) in 2 mM external $Ca^{2+}$ with the indicated stimulation protocol repeated three times. (E) Quantification of P531/P530 for the indicated genotypes (1$^{st}$ stimulation protocol: Control + DMSO, 0.98 ± 0.056, n = 17; Control + bafilomycin, 1.53 ± 0.12, n = 17; *Syt7$^{M1}$* + DMSO, 1.83 ± 0.058, n = 17; *Syt7$^{M1}$* + bafilomycin, 2.10 ± 0.11, n = 17; 2$^{nd}$ stimulation protocol: Control + DMSO, 0.97 ± 0.045, n = 17; Control + bafilomycin, 1.25 ± 0.064, n = 17; *Syt7$^{M1}$* + DMSO, 1.95 ± 0.10, n = 17; *Syt7$^{M1}$* + bafilomycin, 2.09 ± 0.19, n = 17). Statistical significance was determined with a one-way Anova with Sidak's multiple comparisons test. (F) Quantification of mean eEJC amplitudes for P530 and P531 for the indicated genotypes (1$^{st}$ stimulation protocol: P530 in Control + DMSO, 87.39 ± 3.85, n = 17; P531 in Control + DMSO, 80.22 ± 5.25, n = 17; P530 in Control + bafilomycin, 44.68 ± 2.80, n = 17; P531 in Control + bafilomycin, 66.26 ± 5.03, n = 17; P530 in *Syt7$^{M1}$* + DMSO, 97.62 ± 4.04, n = 17; P531 in *Syt7$^{M1}$* + DMSO, 177.34 ± 7.80, n = 17; P530 in *Syt7$^{M1}$* + bafilomycin, 52.44 ± 3.83, n = 17; P531 in *Syt7$^{M1}$* + bafilomycin, 102.50 ± 8.07, n = 17; 2$^{nd}$ stimulation protocol: P530 in Control + DMSO, 68.21 ± 3.97, n = 17; P531 in Control + DMSO, 70.05 ± 5.95, n = 17; P530 in Control + bafilomycin, 15.09 ± 1.26, n = 17; P531 in Control + bafilomycin, 18.15 ± 1.34, n = 17; P530 in *Syt7$^{M1}$* + DMSO, 82.89 ± 4.64, n = 17; P531 in *Syt7$^{M1}$* + DMSO, 163.52 ± 9.74, n = 17; P530 in *Syt7$^{M1}$* + bafilomycin, 11.98 ± 1.26, n = 17; P531 in *Syt7$^{M1}$* + bafilomycin, 24.71 ± 3.00, n = 17). Statistical significance was determined with a Student's paired t-test.

The online version of this article includes the following figure supplement(s) for figure 10:

**Figure supplement 1.** Enhanced recovery after termination of 10 Hz stimulation in *Syt7* mutants.

**Figure supplement 2.** SYT7 overexpression reduces RRP refilling following 10 Hz stimulation.

was introduced at the 3'-end of the endogenous *Syt7* locus using CRISPR (*Figure 11A*). This approach generated a SYT7$^{RFP}$ C-terminal fusion protein expressed under its endogenous enhancers to avoid any overexpression that might trigger changes in its normal localization. The RFP C-terminal fusion did not abolish SYT7 function, as eEJC amplitude in 2 mM $Ca^{2+}$ was not significantly different between control and SYT7$^{RFP}$ (control: 198.9 ± 8.8 nA, n = 14; SYT7$^{RFP}$, 227 ± 11.3 nA, n = 14, p=0.1). A sfGFP version (SYT7$^{GFP}$) was also generated with CRISPR that showed the same intra-terminal expression pattern as SYT7$^{RFP}$ (*Figure 11—figure supplement 1A*). Western blot analysis with anti-RFP identified a single band at the predicted molecular weight (73kD) of the fusion protein in SYT7$^{RFP}$ animals (*Figure 11B*), indicating a single SYT7 isoform is expressed in *Drosophila*. Immunostaining of 3$^{rd}$ instar larvae with anti-RFP antisera revealed SYT7$^{RFP}$ was enriched in presynaptic terminals and formed an expansive tubular network near the plasma membrane that extended into the center of the bouton (*Figure 11C,D*). Neuronal knockdown of *Syt7* with two independent RNAi lines (*elav$^{C155}$*-GAL4; UAS-*Syt7* RNAi) dramatically reduced SYT7$^{RFP}$ on western blots (*Figure 11B*) and eliminated expression of SYT7$^{RFP}$ at the NMJ (*Figure 11—figure supplement 2*), indicating the signal is specific to SYT7 and localizes predominantly to the presynaptic compartment.

To further characterize the subsynaptic localization of SYT7, fluorescently-tagged compartmental markers or compartment-specific antisera were used for labeling in the Syt7$^{RFP}$ background. Images were collected on a Zeiss Airyscan and analyzed in FIJI and Matlab to generate cytofluorogram co-localization plots to calculate the Pearson correlation (r) between SYT7$^{RFP}$ and labeled compartments from individual synaptic boutons at muscle 6/7 NMJs (*Figure 12*, n = 3 animals each). Co-labeling of the SV proteins nSYB and SYT1 served as a positive control (*Figure 12A*, r = 0.71). SYT7$^{RFP}$ and the Golgi marker, Golgin84, served as a negative control since Golgi is absent from presynaptic terminals (*Figure 12L*, r = −0.43). Co-localization analysis indicates SYT7 resides on a membrane compartment that does not completely overlap with any protein tested (*Figure 12B–L*). The largest overlap was with Dynamin (*Figure 12B*, r = 0.22), a GTPase involved in endocytosis that localizes to the peri-AZ. The t-SNARE SYX1 also overlapped with a subset of SYT7 immunolabeling near the plasma membrane (*Figure 12C*, r = 0.15). Although SYT7's pattern of inter-connectivity within the bouton appeared similar to peripheral ER, it did not co-localize with Reticulon-like 1 (RTLN1, *Figure 12D*, r = 0.01), a peripheral ER marker. In addition, SYT7 did not co-localize with SVs (r = −0.17), DCVs labeled with ANF-GFP (r = −0.07), exosomes (r = −0.19), late endosomes (r = −0.29), lysosomes (r = −0.01) or the plasma membrane (anti-HRP, r = −0.06). Neither SYT7$^{RFP}$ (*Figure 12G*, r = −0.11) or SYT7$^{GFP}$ (*Figure 11—figure supplement 1B*) was enriched at AZs, but instead surrounded BRP as previously described for other peri-AZ proteins. These data are in agreement with anti-SYT7 antibody labeling of sucrose gradient-separated subcellular fractions from wildtype *Drosophila* head extracts that localized SYT7 to a distinct membrane compartment separate from SVs and the plasma membrane (*Adolfsen et al., 2004*). In conclusion, SYT7 surrounds AZs marked by BRP (*Figure 11—figure supplement 1B*, *Figure 12G*), indicating the protein localizes in part to the previously described peri-AZ domain. Peri-AZs are enriched in proteins regulating SV

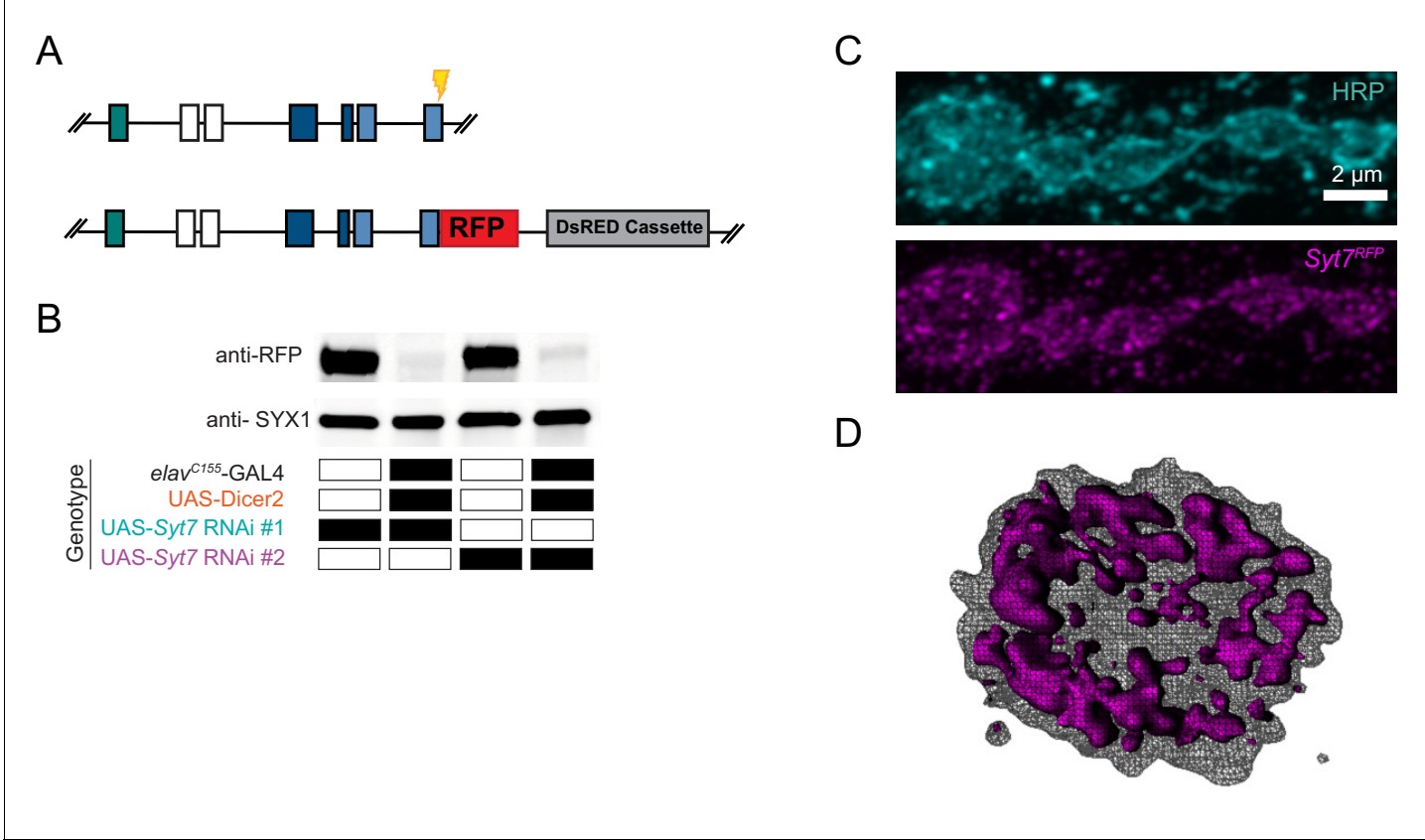

**Figure 11.** Tagging and location of endogenous SYT7. (A) CRISPR strategy used to insert RFP in frame at the *Syt7* 3'end to generate SYT7[RFP]. Exon coloring is the same as *Figure 1D*. The guide RNA cleavage site is displayed in yellow. (B) Two *Syt7* UAS-RNAi lines (#1 and #2) were used to pan-neuronally knockdown SYT7[RFP]. Western blot analysis of head extracts probed with anti-RFP (top panel) from SYT7[RFP] adults following pan-neuronal knockdown of SYT7: lane 1: UAS-*Syt7* RNAi#1; SYT7[RFP]; lane 2: *elav*[C155]-GAL4, UAS-Dicer2; UAS-*Syt7* RNAi#1; SYT7[RFP]; lane 3: UAS-*Syt7* RNAi#2; SYT7[RFP]; lane 4: *elav*[C155]-GAL4, UAS-Dicer2; UAS-*Syt7* RNAi line#2; SYT7[RFP]. SYX1 antisera was used as a loading control (bottom panel). (C) Immunocytochemistry with anti-HRP (top) and anti-RFP (bottom) in SYT7[RFP]3[rd] instar larvae at muscle 6/7 NMJs. SYT7[RFP] staining is abundant in the presynaptic terminal, with a few postsynaptic membrane compartments also labeled. (D) 3D rendering of the terminal bouton (left) from above. The SYT7[RFP] intra-terminal compartment is labeled in magenta, with HRP-labeled plasma membrane indicated with a grey mesh. Scale bar = 2 μm. The online version of this article includes the following figure supplement(s) for figure 11:

**Figure supplement 1.** Location of SYT7[GFP] within synaptic boutons.

**Figure supplement 2.** Knockdown of SYT7[RFP] with *Syt7* RNAi eliminates RFP immunostaining.

endocytosis and endosomal trafficking (*Coyle et al., 2004*; *Koh et al., 2004*; *Marie et al., 2004*; *Rodal et al., 2008*; *Sone et al., 2000*), indicating SYT7 may modulate SV re-entry into the RRP by interfacing with sorting machinery within this domain.

SYT7 localization was widespread within the peri-AZ region, with SYT7[RFP] tubules in close proximity to other labeled membrane compartments, including endosomes, lysosomes, and the plasma membrane (*Figure 12—figure supplement 1*). To determine if the SYT7 compartment required endosomal trafficking for its assembly or maintenance, a panel of dominant-negative, constitutively-active or wildtype endosomal UAS-RAB proteins (*Zhang et al., 2007*) were expressed with *elav*[C155]-GAL4 in the SYT7[RFP] background. Manipulations of RAB5 (early endosomes), RAB7 (late endosomes) or RAB4 and RAB11 (recycling endosomes) did not disrupt the abundance or morphology of the SYT7 tubular network (*Figure 12—figure supplement 2*). Similarly, no change in the distribution of several compartment markers were found in *Syt7*[M1] mutants, including the early endosomal marker RAB5, the late endosomal/peri-AZ marker RAB11 and the peri-AZ protein Nervous Wreck (NWK) (*Figure 12—figure supplement 3*). In addition, no defect was observed in trans-synaptic transfer of the exosomal protein SYT4 to the postsynaptic compartment, indicating SYT7 does not regulate

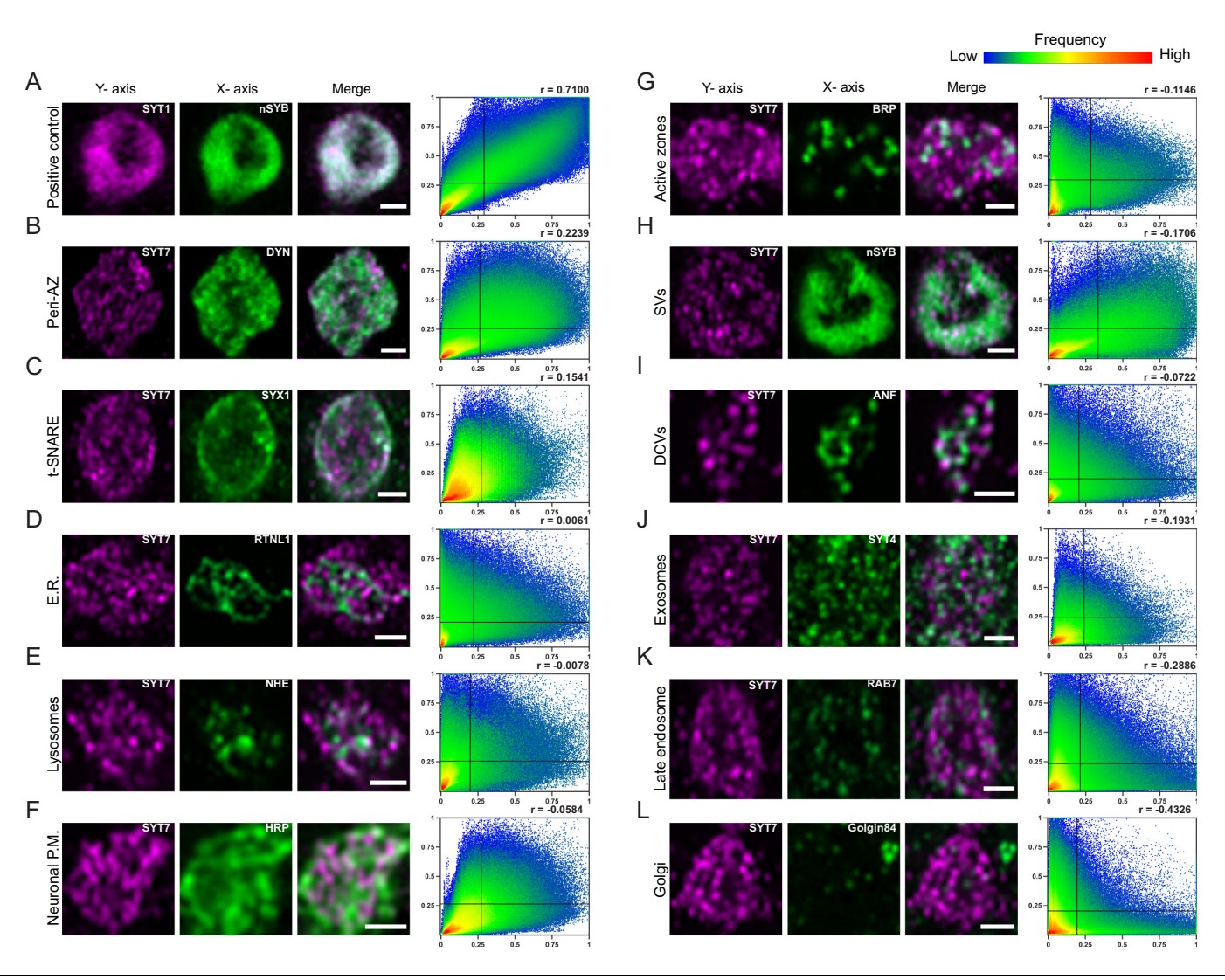

**Figure 12.** Localization of SYT7 in presynaptic terminals. Immunostaining for the indicated proteins in each panel was performed at 3rd instar larval muscle 6/7 NMJs. Staining for all panels except A were done in the SYT7$^{RFP}$ endogenously tagged background using anti-RFP to label the SYT7 compartment, with the merged image shown on the right. The Pearson correlation coefficient (r) calculated from the cytofluorogram co-localization plots is shown on the upper right. All images are from single confocal planes. (**A**) Co-localization of the SV proteins SYT1 (left, magenta, anti-SYT1 antisera) and nSYB (middle, green, endogenous nSYB$^{GFP}$) as a positive control. The remaining panels show boutons co-stained for SYT7$^{RFP}$ (left, magenta, anti-RFP antisera) and the indicated compartment marker (middle, green): (**B**) Dynamin (anti-DYN antisera); (**C**) SYX1 (anti-SYX1 antisera); (**D**) Reticulin like-1 (*elav*$^{C155}$-GAL4; UAS-RTNL1-GFP); (**E**) lysosomal Na$^+$/H$^+$ exchanger 1 (*elav*$^{C155}$-GAL4; UAS-NHE-GFP); (**F**) HRP (anti-HRP antisera); (**G**) BRP (anti-BRP Nc82 antisera); (**H**) nSYB (nSYB$^{GFP}$); (**I**) Atrial natriuretic peptide (*elav*$^{C155}$-GAL4; UAS-ANF-GFP); (**J**) SYT4 (endogenously tagged SYT4$^{GFP-2M}$); (**K**) RAB7 (anti-RAB7 antisera); and (**L**) Golgin84 (anti-GOLGIN84 antisera). Co-localization plots were generated with normalized pixel intensity of stacked images of 10–24 type Ib boutons from three animals per genotype, with the color representing the frequency of data points as shown in the right scale bar. The vertical line on the X-axis indicates the threshold used to identify pixels above background for the compartment stain. The horizontal line on the Y-axis represents the threshold used to identify pixels above background for SYT7. Scale bar = 1 μm.

The online version of this article includes the following figure supplement(s) for figure 12:

**Figure supplement 1.** SYT7 tubules reside in proximity to multiple presynaptic compartments.
**Figure supplement 2.** SYT7 localization is not altered by specific RAB protein manipulations.
**Figure supplement 3.** Localization of compartment-specific markers in *Syt7* mutants.
**Figure supplement 4.** Model for SYT7 localization and function.

exosome trafficking as described for several other peri-AZ proteins (*Walsh et al., 2019*). Although no sub-compartment overlapped completely with SYT7, the protein is positioned within the peri-AZ to interact with SVs, endosomes and the recycling machinery to negatively regulate the size of releasable SV pools (*Figure 12—figure supplement 4*). We conclude that SYT7 does not localize to SVs and is not enriched at AZs, consistent with SYT7 negatively regulating SV release through an indirect mechanism that does not require its presence at sites of SV fusion.

## Discussion

To characterize the location and function of SYT7 in *Drosophila* we used the CRISPR-Cas9 system to endogenously label the protein and generate null mutations in the *Syt7* locus. Our findings indicate SYT7 acts as a negative regulator of SV release, AZ $P_r$, RRP size, and RRP refilling. The elevated $P_r$ across the AZ population in *Syt7* mutants provides a robust explanation for why defects in asynchronous release and facilitation are observed in the absence of the protein, as SYT7 levels set the baseline for the amount of evoked release. SYT7's presence on an internal tubular membrane network within the peri-AZ positions the protein to interface with the SV cycle at multiple points to regulate membrane trafficking. In addition, increased SV release in animals lacking both SYT1 and SYT7 indicate the full complement of $Ca^{2+}$ sensors that mediate the distinct phases of SV release remain unknown.

### Syt7 mutants have increased $P_r$ at *Drosophila* NMJs

Using a combination of synaptic physiology and imaging approaches, our findings indicate SYT7 acts to reduce SV recruitment and release. Minor defects in asynchronous release and facilitation were identified in *Drosophila Syt7* mutants, as observed in mouse and zebrafish models (*Bacaj et al., 2013*; *Chen et al., 2017*; *Jackman et al., 2016*; *Turecek and Regehr, 2019*; *Turecek and Regehr, 2018*; *Weber et al., 2014*; *Wen et al., 2010*). However, we attribute these defects to reduced SV availability as a result of increased $P_r$ in *Syt7* mutants. Indeed, a key feature of facilitation is its critical dependence on initial $P_r$ (*Neher and Brose, 2018*; *Zucker and Regehr, 2002*). Low $P_r$ synapses increase SV fusogenicity as $Ca^{2+}$ levels rise during paired-pulses or stimulation trains, resulting in short-term increases in $P_r$ for SVs not recruited during the initial stimuli. In contrast, depression occurs at high $P_r$ synapses due to the rapid depletion of fusion-capable SVs during the initial response. Prior quantal imaging at *Drosophila* NMJs demonstrated facilitation and depression can occur across different AZs within the same neuron, with high $P_r$ AZs depressing and low $P_r$ AZs facilitating (*Peled and Isacoff, 2011*). Given the elevated $P_r$ in *Syt7* mutants, the facilitation defects are likely related to differences in initial $P_r$ and depletion of fusion-competent SVs available for release during the 2nd stimuli.

A similar loss of SVs due to elevated $P_r$ in *Syt7* mutants would reduce fusogenic SVs that are available during the delayed phase of the asynchronous response. *Syt1; Syt7* double mutants continue to show asynchronous fusion and facilitation, demonstrating there must be other $Ca^{2+}$ sensors that mediate these components of SV release. The predominant localization of endogenous SYT7 to an internal tubular membrane compartment at the peri-AZ also places the majority of the protein away from release sites where it would need to reside to directly activate SV fusion. As such, our data indicate SYT7 regulates SV release through a distinct mechanism from SYT1.

We can also conclude that the remaining components of asynchronous fusion and facilitation must be mediated by an entirely different family of $Ca^{2+}$-binding proteins than Synaptotagmins (or through $Ca^{2+}$-lipid interactions). Of the seven *Syt* genes in the *Drosophila* genome, only 3 SYT proteins are expressed at the motor neuron synapses assayed in our study – SYT1, SYT4 and SYT7 (*Adolfsen et al., 2004*). For the remaining SYTs in the genome, SYT-α and SYT-β are expressed in neurosecretory neurons and function in DCV fusion (*Adolfsen et al., 2004*; *Park et al., 2014*). SYT12 and SYT14 lack $Ca^{2+}$ binding residues in their C2 domains and are not expressed in motor neurons (*Adolfsen et al., 2004*). In addition, SYT4 is found on exosomes and transferred to postsynaptic cells, where it participates in retrograde signaling (*Adolfsen et al., 2004*; *Harris et al., 2016*; *Korkut et al., 2013*; *Walsh et al., 2019*; *Yoshihara et al., 2005*). *Syt1; Syt4* double mutants display the same SV fusion defects found in *Syt1* mutants alone, indicating SYT4 cannot compensate for SYT1 function in SV release (*Barber et al., 2009*; *Saraswati et al., 2007*). As such, SYT1 and SYT7

are the only remaining SYT isoforms that could contribute to SV trafficking within *Drosophila* motor neuron terminals.

A prior study from our lab using a *Syt7* exon-intron hairpin RNAi we generated did not result in an increase in evoked release (*Saraswati et al., 2007*). Although a reduction in ectopic expression of SYT7 in muscles could be seen with *Mhc*-GAL4 driving the UAS-*Syt7* RNAi, our anti-SYT7 antisera does not recognize the endogenous protein in neurons using immunocytochemistry, preventing a determination of presynaptic SYT7 levels following neuronal RNAi. To further examine this issue, we performed western blot analysis with this RNAi and compared it those used in the current study. Our results confirmed that the RNAi line failed to reduce endogenous GFP-tagged SYT7 (data not shown), although the two commercial RNAi lines we used here were highly effective (*Figure 11B*). Based on these data, we conclude that the previous *Syt7* UAS-RNAi line was ineffective in knocking down endogenous SYT7. Given the *Syt7^{M1}* and *Syt7^{M2}* alleles result in early stop codons and lack SYT7 expression by western blot analysis and display elevated levels of fusion, our data indicate SYT7 normally acts to suppress SV release as demonstrated by electrophysiology and optical $P_r$ imaging. SYT7 overexpression reduces SV release even more, further confirming that the levels of SYT7 set the baseline amount of SV fusion at *Drosophila* NMJ synapses.

## SYT7 regulates the recruitment and fusion of SVs in a dose-dependent manner

Although our data indicate SYT7 is not the primary asynchronous or facilitation $Ca^{2+}$ sensor in *Drosophila,* we found it inhibits SV release in a dosage-sensitive manner. The reduction in SV release is not due to changes in the $Ca^{2+}$ cooperativity of fusion or enhanced presynaptic $Ca^{2+}$ entry, ruling out the possibility that SYT7 normally acts as a local $Ca^{2+}$ buffer or an inhibitor of presynaptic voltage-gated $Ca^{2+}$ channels. The reduction in release is also not due to altered endocytosis, as *Syt7* mutants have a normal steady-state rate of SV cycling following depletion of the RRP. Instead, SYT7 regulates SV fusogenicity at a stage between SV endocytosis and fusion. Given the rapid enhanced refilling of the RRP observed in *Syt7* mutants, and the suppression of RRP refilling following SYT7 overexpression, our data indicate SYT7 regulates releasable SVs in part through changes in SV mobilization to the RRP. $Ca^{2+}$ is well known to control the replenishment rate of releasable SVs, with Calmodulin-UNC13 identified as one of several molecular pathways that accelerate RRP refilling in a $Ca^{2+}$-dependent manner (*Dittman et al., 2000*; *Dittman and Regehr, 1998*; *Junge et al., 2004*; *Lipstein et al., 2013*; *Ritzau-Jost et al., 2018*). Our findings indicate SYT7 acts in an opposite fashion and slows RRP refilling, providing a $Ca^{2+}$-dependent counter-balance for SV recruitment into the RRP. Although such an effect has not been described for mammalian SYT7, defects in RRP replenishment have been observed when both SYT1 and SYT7 are deleted or following high frequency stimulation trains (*Bacaj et al., 2015*; *Durán et al., 2018*; *Liu et al., 2014*).

SYT7's role in restricting SV fusion and RRP size also affects spontaneous release. Although *Syt7* mutants alone do not show elevated mini frequency, *Double^{Null}* mutants have a 2-fold increase in spontaneous release. Similar increases in spontaneous release were observed at mammalian synapses lacking both SYT7 and SYT1 (or SYT2), with the effect being attributed to a dual role in clamping fusion in the absence of $Ca^{2+}$ (*Luo and Südhof, 2017*; *Turecek and Regehr, 2019*). Our results indicate the elevation in spontaneous release at *Drosophila* synapses is a result of an increase in releasable SVs rather than a clamping function for SYT7. Following overexpression of SYT7, there is a reduction in the number of fusogenic SVs available for both evoked and spontaneous release. The dosage-sensitivity of the various phenotypes indicate SYT7 abundance is a critical node in controlling SV release rate. Indeed, mammalian SYT7 levels are post-transcriptionally modulated by γ-secretase proteolytic activity and APP, linking it to SV trafficking defects in Alzheimer's disease (*Barthet et al., 2018*).

## How does SYT7 negatively regulate recruitment and fusion of SVs?

The precise mechanism by which SYT7 reduces release and slows refilling of the RRP is uncertain given it is not enriched at sites of SV fusion. Although we cannot rule out the possibility that a small fraction of the protein is found at AZs, SYT7 is predominantly localized to an internal membrane compartment at the peri-AZ where SV endocytosis and endosomal sorting occurs (*Coyle et al., 2004*; *Koh et al., 2004*; *Marie et al., 2004*; *Rodal et al., 2008*; *Sone et al., 2000*). SYT7 membrane

tubules are in close proximity and could potentially interact with peri-AZs proteins, endosomes, lysosomes and the plasma membrane. Given its primary biochemical activity is to bind membranes in a $Ca^{2+}$-dependent manner, SYT7 could mediate cargo or lipid movement across multiple compartments within peri-AZs. In addition, it is possible SYT7 tubules could form part of the poorly defined SV recycling endosome compartment. However, we observed no change in SV density or SV localization around AZs, making it unlikely SYT7 would be essential for endosomal trafficking of SVs. The best characterized regulator of the SV endosome compartment in *Drosophila* is the RAB35 GAP Skywalker (SKY) (*Uytterhoeven et al., 2011*). Although *Sky* mutations display some similarities to *Syt7*, including increased neurotransmitter release and larger RRP size, *Syt7* lacks most of the well-described *Sky* phenotypes such as behavioral paralysis, FM1-43 uptake into discrete punctated compartments, cisternal accumulation within terminals and reduced SV density. In addition, we found no co-localization between SKY-GFP and SYT7[RFP] within presynaptic terminals.

By blocking SV refilling with bafilomycin, our findings indicate the fast recovery of the RRP can occur via enhanced recruitment from the reserve pool and does not require changes in endocytosis rate. The phosphoprotein Synapsin has been found to maintain the reserve SV pool by tethering vesicles to actin filaments at rest (*Akbergenova and Bykhovskaia, 2007*; *Bykhovskaia, 2011*; *Hilfiker et al., 1999*; *Milovanovic and De Camilli, 2017*; *Shupliakov et al., 2011*). Synapsin interacts with the peri-AZ protein Dap160/Intersectin to form a protein network within the peri-AZ that regulates clustering and release of SVs (*Gerth et al., 2017*; *Marie et al., 2004*; *Winther et al., 2015*). Synapsin-mediated phase separation is also implicated in clustering SVs near release sites (*Milovanovic et al., 2018*; *Milovanovic and De Camilli, 2017*). SYT7 could similarly maintain a subset of SVs in a non-releasable pool and provide a dual mechanism for modulating SV mobilization. Phosphorylation of Synapsin and $Ca^{2+}$ activation of SYT7 would allow multiple activity-dependent signals to regulate SV entry into the RRP. As such, SYT7 could play a key role in organizing membrane trafficking and protein interactions within the peri-AZ network by adding a $Ca^{2+}$-dependent regulator of SV recruitment and fusogenicity.

Additional support for a role for SYT7 in regulating SV availability through differential SV sorting comes from recent studies on the SNARE complex binding protein CPX. Analysis of *Drosophila Cpx* mutants, which have a dramatic increase in minis (*Buhl et al., 2013*; *Huntwork and Littleton, 2007*; *Jorquera et al., 2012*), revealed a segregation of recycling pathways for SVs undergoing spontaneous versus evoked fusion (*Sabeva et al., 2017*). Under conditions where intracellular $Ca^{2+}$ is low and SYT7 is not activated, spontaneously-released SVs do not transit to the reserve pool and rapidly return to the AZ for re-release. In contrast, SVs released during high frequency evoked stimulation when $Ca^{2+}$ is elevated and SYT7 is engaged, re-enter the RRP at a much slower rate. This mechanism slows re-entry of SVs back into the releasable pool when stimulation rates are high and large numbers of SV proteins are deposited onto the plasma membrane at the same time, allowing time for endosomal sorting that might be required in these conditions. In contrast, SVs released during spontaneous fusion or at low stimulation rates would likely have less need for endosomal re-sorting. Given SYT7 restricts SV transit into the RRP, it provides an activity-regulated $Ca^{2+}$-triggered switch for redirecting and retaining SVs in a non-fusogenic pool that could facilitate sorting mechanisms.

Beyond SV fusion, presynaptic membrane trafficking is required for multiple signaling pathways important for developmental maturation of NMJs (*Harris and Littleton, 2015*; *McCabe et al., 2003*; *Packard et al., 2002*; *Piccioli and Littleton, 2014*; *Rodal et al., 2011*). In addition, alterations in neuronal activity or SV endocytosis can result in synaptic undergrowth or overgrowth (*Akbergenova et al., 2018*; *Budnik et al., 1990*; *Dickman et al., 2006*; *Guan et al., 2005*; *Koh et al., 2004*). We did not find any defect in synaptic bouton or AZ number, indicating SYT7 does not participate in membrane trafficking pathways that regulate synaptic growth and maturation. However, a decrease in T-bar area and presynaptic $Ca^{2+}$ influx in *Syt7* mutants was found. Although it is unclear how these phenotype arise, they may represent a form of homeostatic plasticity downstream of elevated synaptic transmission (*Frank et al., 2020*). There is also ample evidence that SV distance to $Ca^{2+}$ channels plays a key role in defining the kinetics of SV release and the size of the RRP (*Böhme et al., 2016*; *Chen et al., 2015*; *Neher, 2015*; *Neher and Brose, 2018*; *Wadel et al., 2007*), suggesting a change in such coupling in *Syt7* mutants might contribute to elevations in $P_r$ and RRP refilling. Further studies will be required to precisely define how SYT7 controls the baseline level of SV release at synapses.

# Materials and methods

## *Drosophila* stocks

*Drosophila melanogaster* were cultured on standard medium at 22–25°C. Genotypes used in the study include: elav^C155^-GAL4 (Bloomington *Drosophila* Stock Center (BDSC)#8765), UAS-ANF-Emerald (BDSC#7001), SYT4^GFP-2M^ (*Harris et al., 2016*), Syt1^AD4^ (*DiAntonio and Schwarz, 1994*), Syt1^N13^ (*Littleton et al., 1993b*), UAS-Syt7 (*Saraswati et al., 2007*), *Mhc*-GAL4 (BDSC#55132), UAS-Syt7 RNAi#1 (Vienna#24989) and UAS-Syt7 RNAi#2 (BDSC#27279). Lines used for testing co-localization with SYT7^RFP^ or mis-localization in Syt7^M1^ include: endogenous nSYB^GFP^ (this study), UAS-NHE-GFP (this study), UAS-ANF-Emerald (BDSC#7001), SYT4^GFP-2M^ (*Harris et al., 2016*), UAS-RTNL1-GFP (BDSC#77908), RAB5-YFP (BDSC#62543) and RAB11-YFP (BDSC#62549). Lines used for assaying SYT7^RFP^ localization after overexpressing RABs: UAS-RAB4-YFP (BDSC#9767), UAS-RAB4 (Q67L)-YFP (BDSC#9770), UAS-RAB5-YFP (BDSC#24616), UAS-RAB5(S43N)-YFP (BDSC#9772), UAS-RAB5(T22N)-YFP (BDSC#9778), UAS-RAB7-YFP (BDSC#23641), UAS-RAB7(Q67L)-YFP (BDSC#9779), UAS-RAB11-YFP (BDSC#50782) and UAS-RAB11(S25N)-YFP (BDSC#9792) (*Zhang et al., 2007*).

## Genome engineering of Syt7^M1^ mutant and SYT7^RFP^ knock-in

Guide RNAs were selected using the CRISPR Optimal Target Finder resource (*Gratz et al., 2014*) and cloned into the plasmid pCFD4-U6:1_U6:3tandemgRNAs (Addgene #49411) (*Port et al., 2014*). To generate *Syt7^M1^*, guide RNA containing pCFD4 plasmid was inject into *vasa*-Cas9 embryos (BDSC #56552) by Best Gene Inc (Chino Hills, CA, USA). *Syt7^M1^* and an unaffected injection line (control) were brought into the *white* background and the *vasa*-Cas9 chromosome was removed. To generate SYT7^RFP^, a donor plasmid that flanked RFP and a DsRed cassette was generated from the pScarless plasmid (courtesy of Kate O'Connor-Giles) with 1 Kb homology arms from the 3' end of the *Syt7* gene. The left homology arm was generated by PCR and the right homology arm was synthesized by Epoch Life Science (Sugarland, TX, USA). The donor plasmid and guide RNA containing pCFD4 plasmid was co-injected into Act5C-Cas9, Lig4 (BDSC #58492) by Best Gene Inc *Syt7^M1^* and SYT7^RFP^ transformants were confirmed by DNA sequencing.

## Sequence alignment, phylogenetic tree construction and molecular modeling

NCBI BLAST was used to identify homologs of SYT1, SYT7 and ESYT-2 in the genomes of *C. elegans*, *C. intestinalis*, *D. rerio*, *M. musculus*, *H. sapiens*, *R. norvegicus* and *T. adherens*. Jalview was used to align SYT1 and SYT7 protein sequences from *D. melanogaster*, *M. Musculus* and *H. sapiens* with the T-coffee multiple sequence alignment algorithm. Jalview and Matlab were used to generate a phylogenetic tree using BLOSUM62 matrix and neighbor joining clustering. The SWISS model server (https://swissmodel.expasy.org) was used for homology modeling of *Drosophila* SYT7 from *R. norvegicus* SYT7 (PBD: 6ANK) (*Waterhouse et al., 2018*). The PyMOL Molecular Graphics System (Version 2.0 Schrödinger, LLC) was used to visualize SYT1 and SYT7 protein structures.

**Sequences used for sequence alignment and phylogenetic tree**

| Protein | Species | NCBI accession number |
|---|---|---|
| ESYT2 | *C. elegans* | NP_741181.1 |
| | *C. intestinalis* | XP_018671537.1 |
| | *D. melanogaster* | NP_733011.2 |
| | *D. rerio* | XP_005171456.1 |
| | *H. sapiens* | XP_024302614.1 |
| | *R. norvegicus* | NP_001258098.1 |
| | *T. adhaerens* | EDV19885.1 |

*Continued on next page*

*Continued*

**Sequences used for sequence alignment and phylogenetic tree**

| SYT1 | | |
|---|---|---|
| | *C. elegans* | NP_495394.3 |
| | *C. intestinalis* | NP_001107602.1 |
| | *D. melanogaster* | NP_523460.2 |
| | *D. rerio* | NP_001314758.1 |
| | *H. sapiens* | NP_001129277.1 |
| | *R. norvegicus* | NP_001028852.2 |
| | *T. adhaerens* | XP_002117742.1 |
| SYT7 | | |
| | *C. elegans* | NP_001254022.1 |
| | *C. intestinalis* | XP_026696415.1 |
| | *D. melanogaster* | NP_726560.5 |
| | *D. rerio* | XP_021326273.1 |
| | *H. sapiens* | NP_004191.2 |
| | *R. norvegicus* | NP_067691.1 |
| | *T. adhaerens* | XP_002117784.1 |

## Western blot analysis and immunocytochemistry

Western blotting of adult head lysates (one head/lane) was performed using standard laboratory procedures with anti-SYT7 (1:500) (*Adolfsen et al., 2004*), anti-SYX1 (8C3, 1:1000, Developmental Studies Hybridoma Bank (DSHB, Iowa City, IA) and anti-RFP (600-401-379; Rockland, 1:5000). Visualization and quantification were performed with a LI-COR Odyssey Imaging System (LI-COR Biosciences, Lincoln, MA, USA). Secondary antibodies for western blotting included Alexa Fluor 680-conjugated goat anti-rabbit IgG (1:5000, Invitrogen; A21109) and IR Dye 800-conjugated goat anti-mouse IgG (1:5000, LICOR; 926–32211).

Immunostaining for AZ and bouton counting was performed on wandering stage 3rd instar larvae dissected in $Ca^{2+}$-free HL3.1 and fixed for 17 min in $Ca^{2+}$-free HL3.1 containing 4% PFA. Larvae were blocked and permeabilized for 1 hr in PBS containing 0.1% Triton X-100, 2.5% NGS, 2.5% BSA and 0.1% sodium azide. Larvae were incubated overnight with primary antibody at 4°C and 2 hr in secondary antibody at room temperature. Samples were mounted on slides with Vectashield (Vector Laboratories, Burlingame, CA). Immunostaining for SYT7^RFP and STY7^GFP co-localization analysis was similar, except larvae were blocked and permeabilized overnight in PBS containing 0.25% Saponin, 2.5% normal goat serum (NGS), 2.5% bovine serum albumin (BSA) and 0.1% sodium azide. Fixed larvae were incubated with primary antibody at 4°C for 24 hr and with secondary antibodies for 1.5 hr at room temperature. Fixed larvae were mounted in ProLong Diamond Antifade Mountant (#P36970; Thermo Fisher Scientific, Waltham, MA, USA).

Antibodies used for immunolabeling were: mouse anti-BRP at 1:500 (Nc82; DSHB), mouse anti-DYN at 1:1000 (Clone 41, Dynamin (RUO); BD Transduction Laboratories, San Jose, CA, USA), mouse anti-Golgin84 at 1:50 (Golgin84 12–1; DSHB), mouse anti-RAB7 at 1:10 (Rab7; DSHB), mouse anti-RFP at 1:1000 (200-301-379; Rockland, Limerick, PA, USA) mouse anti-SYX1 at 1:100 (8C3; DSHB), rabbit anti-CPX at 1:5000 (*Huntwork and Littleton, 2007*), rabbit anti-NWK at 1:1000 (gift from Avital Rodal), rabbit anti-SYT1 1:500, mouse anti-GFP at 1:1000 (#A-11120; Thermo Fisher Scientific, Waltham, MA, USA), rabbit anti-GFP at 1:1000 (#G10362; Thermo Fisher Scientific, Waltham, MA, USA), mouse anti-RFP at 1:1000 (200-301-379; Rockland), rabbit anti-RFP at 1:1000 (600-401-379; Rockland) and DyLight 649 conjugated anti-HRP at 1:1000 (#123-605-021; Jackson Immuno Research, West Grove, PA, USA). Secondary antibodies used for AZ and bouton counting were used at 1:1000: goat anti-rabbit Alexa Flour 488-conjugated antibody (A-11008; Thermofisher) and goat anti-mouse Alexa Fluor 546-conjugated antibody (A-11030; ThermoFisher). Secondary antibodies used for co-localization were used at 1:1000: goat anti-mouse Alexa Fluor Plus 555 (A32727; Thermofisher), goat anti-mouse Alexa Fluor Plus 488 (A32723; ThermoFisher), goat anti-rabbit Alexa

Fluor Plus 555 (A32732; ThermoFisher) and goat anti-rabbit Alexa Fluor Plus 488 (A32731; ThermoFisher).

Immunoreactive proteins were imaged on either a Zeiss Pascal Confocal (Carl Zeiss Microscopy, Jena, GERMANY) using a 40x or 63X NA 1.3 Plan Neofluar oil immersion objective or a ZEISS LSM 800 microscope with Airyscan using a 63X oil immersion objective. For AZ volume and AZ proximity measurements, samples were imaged on a Zeiss Airyscan microscope and BRP labeling was analyzed in Volocity 6.3.1 software (Quorum Technologies Inc, Puslinch, Ontario, CAN). AZs clusters larger than 0.2 $\mu m^3$ were rarely found, but could not be resolved into single objects by the software. To ensure such clusters did not affect AZ size analysis, all AZs larger than 0.2 $\mu m^3$ were excluded from the analysis.

## Electrophysiology

Postsynaptic currents from the indicated genotypes were recorded from $3^{rd}$ instar muscle fiber 6 at segment A3 using two-electrode voltage clamp with a $-80$ mV holding potential in HL3.1 saline solution (in mM, 70 NaCl, 5 KCl, 10 NaHCO3, 4 MgCl2, five trehalose, 115 sucrose, 5 HEPES, pH 7.2) as previously described (*Jorquera et al., 2012*). Final $[Ca^{2+}]$ was adjusted to the level indicated in the text. The $Ca^{2+}$ cooperativity of release was determined from the slopes of a linear fit of a double logarithmic plot of evoked responses in the linear range (0.175 to 0.4 mM $Ca^{2+}$). Inward currents recorded during TEVC are labeled as positive values in the figures for simplicity. For experiments using bafilomycin, 4 $\mu m$ bafilomycin (LC Laboratories, Woburn, MA, USA) was dissolved in dimethyl sulphoxide (DMSO, Sigma, St. Louis, MO, USA) in HL3.1 and bath applied to dissected larvae. DMSO containing HL3.1 was used for control. Data acquisition and analysis was performed using Axoscope 9.0 and Clampfit 9.0 software (Molecular Devices, Sunnyvale, CA, USA). mEJCs were analyzed with Mini Analysis software 6.0.3 (Synaptosoft, Decatur, GA, USA). Motor nerves innervating the musculature were severed and placed into a suction electrode. Action potential stimulation was applied at the indicated frequencies using a programmable stimulator (Master8, AMPI; Jerusalem, Israel).

## Fluo-4 AM imaging

Fluo-4 AM (F14201; ThermoFisher) loading was performed as previously described (*Dawson-Scully et al., 2000*). During incubation, neuronal membranes were labeled with DyLight 649 conjugated anti-HRP at 1:1000 (#123-605-021; Jackson Immuno Research, West Grove, PA, USA). NMJs of Ib motoneurons at muscle 6/7 were identified and motor nerves were stimulated in HL3 saline with 20 mM $MgCl_2$ and 1.1 mM $Ca^{2+}$ for 5 s at 10 Hz for three epochs, each with a 5 s rest period between stimulation. Imaging of Fluo-4 AM fluorescent signal was performed on a Zeiss Axio Imager two equipped with a spinning-disk confocal head (CSU-X1; Yokagawa, JAPAN) and ImagEM X2 EM-CCD camera (Hamamatsu, Hamamatsu City JAPAN). 5 $\mu m$ stacks from synaptic boutons were imaged at a frame rate of 1.25 Hz and mean Fluo-4 AM fluorescent intensity was determined during the stimulation protocol for each trial.

## Optical quantal imaging and $P_r$ mapping

$P_r$ mapping was performed on a Zeiss Axio Imager equipped with a spinning-disk confocal head (CSU-X1; Yokagawa, JAPAN) and ImagEM X2 EM-CCD camera (Hamamatsu, Hamamatsu City JAPAN) as previously described (*Akbergenova et al., 2018*). Myristoylated-GCaMP6s was expressed in larval muscles with 44H10-LexAp65 (provided by Gerald Rubin). Individual PSDs were visualized by expression of GluRIIA-RFP under its endogenous promoter (provided by Stephan Sigrist). An Olympus LUMFL N 60X objective with a 1.10 NA was used to acquire GCaMP6s imaging data at 8 Hz. $3^{rd}$ instar larvae were dissected in $Ca^{2+}$-free HL3 containing 20 mM $MgCl_2$. After dissection, preparations were maintained in HL3 with 20 mM $MgCl_2$ and 1.0 mM $Ca^{2+}$ for 5 min. A dual channel multiplane stack was imaged at the beginning of each experiment to identify GluRIIA-positive PSDs. Single focal plane videos were then recorded while motor nerves were stimulated with a suction electrode at 1 Hz. GluRIIA-RFP PSD position was re-imaged every 25 s during experiments. The dual channel stack was merged with single plane images using the max intensity projection algorithm from Volocity 6.3.1 software. The position of all GluRIIA-RFP PSDs was then spliced with the myr-GCaMP6s stimulation video. GluRIIA positive PSDs were detected automatically using the spot

finding function of Volocity and equal size ROIs were assigned to the PSD population. In cases where the software failed to label visible GluRIIA-RFP PSDs, ROIs were added manually. GCaMP6s peak flashes were then detected and assigned to ROIs based on centroid proximity. The time and location of $Ca^{2+}$ events were imported into Excel or Matlab for further analysis. Observed GCaMP events per ROI were divided by stimulation number to calculate AZ $P_r$.

## FM1-43 uptake and release assays

$3^{rd}$ instar wandering larvae were dissected in $Ca^{2+}$-free HL3.1 and axons were severed from the CNS. Axon bundles were stimulated with a suction electrode in 1.5 mM $CaCl_2$ HL3.1 solution containing 2 µM of the lipophilic dye FM 1-43FX (F35355; Thermo Fisher Scientific, Waltham, MA, USA). Dye loading was performed at 10 Hz for 50 s (500 events) or at 0.5 Hz for 300 s (150 events), 600 s (300 events) and 900 s (600 events) as indicated. After stimulation, samples were washed for 2 min in $Ca^{2+}$ free HL3.1 containing 100 µM Advacep-7 (Sigma; A3723) to help remove non-internalized FM 1–43 dye. Image stacks from muscle 6/7 at segment A3 were obtained using a spinning disk confocal microscope. FM1-43 unloading was done with a high $K^+$ (90 mM) HL3.1 solution for 1 min, followed by washing in a $Ca^{2+}$ free HL3.1 solution for 1 min. An image stack at segment A3 muscle 6–7 was obtained on a Zeiss Axio Imager two equipped with a spinning-disk confocal head with a 63X water immersion objective. Mean FM1-43 intensity at the NMJ was quantified using the Volocity 3D Image Analysis software (Quorum Technologies Inc, Puslinch, Ontario, CAN).

## Electron microscopy

$Syt1^{M1}$ and control $3^{rd}$ instar larvae were dissected in $Ca^{2+}$-free HL3.1 solution and fixed in 1% glutaraldehyde, 4% formaldehyde, and 0.1 m sodium cacodylate for 10 min at room temperature as previously described (*Akbergenova and Bykhovskaia, 2009*). Fresh fixative was added and samples were microwaved in a BioWave Pro Pelco (Ted Pella, Inc, Redding, CA, USA) with the following protocol: (1) 100W 1 min, (2) 1 min off, (3) 100W 1 min, (4) 300W 20 secs, (5) 20 secs off, (6) 300W 20 secs. Steps 4–6 were repeated twice more. Samples were then incubated for 30 min at room temperature with fixative. After washing in 0.1 M sodium cacodylate and 0.1 M sucrose, samples were stained for 30 min in 1% osmium tetroxide and 1.5% potassium ferrocyanide in 0.1 M sodium cacodylate solution. After washing with 0.1 M sodium cacodylate, samples were stained for 30 mins in 2% uranyl acetate and dehydrated through a graded series of ethanol and acetone, before embedding in epoxy resin (Embed 812; Electron Microscopy Sciences). Thin sections (50–60 nm) were collected on Formvar/carbon-coated copper slot grids and contrasted with lead citrate. Sections were imaged at 49,000 × magnification at 80 kV with a Tecnai G2 electron microscope (FEI, Hillsboro, OR, USA) equipped with a charge-coupled device camera (Advanced Microscopy Techniques, Woburn, MA, USA). Type Ib boutons at muscle 6/7 were analyzed. All data analyses were done blinded.

For SV counting, T-bars at Ib boutons were identified and a FIJI macro was used to draw four concentric circles with 100 nm, 200 nm, 300 nm or 400 nm radius. The concentric circles were drawn with the T-bar at the center. To quantify vesicle density, FIJI was used to measure the area of the bouton and quantify the total number of vesicles within it. Final analysis was performed in Matlab and Excel.

## Co-localization analysis and 3D reconstruction

The JaCOP FIJI algorithm (*Bolte and Cordelières, 2006*) was used to obtain cytofluorogram plots of bouton image stacks that were probed for RFP and a $2^{nd}$ labeled compartment in SYT7$^{RFP}$$3^{rd}$ instar larvae. Automatic thresholding was used to identify pixels above background for both channels. To obtain an average Pearson correlation, cytofluorograms from boutons obtained from three animals were analyzed in Matlab. All data analyseis were done blinded. 3D reconstruction was performed using the 3D Viewer plugin in FIJI (*Schmid et al., 2010*). The bouton stack was displayed as a surface and labeled with SYT7$^{RFP}$ in magenta and HRP in black.

## Statistical analysis

Statistical analysis and graphing was performed with either Origin Software (OriginLab Corporation, Northampton, MA, USA) or GraphPad Prism (San Diego, CA, USA). Statistical significance was

determined using specific tests as indicated in the text. Appropriate sample size was determined using GraphPad Statmate. Asterisks denote p-values of: *, $p \leq 0.05$; **, $p \leq 0.01$; and ***, $p \leq 0.001$. All histograms and measurements are shown as mean ± SEM.

## Acknowledgements

This work was supported by NIH grant NS40296 to JTL MCQ-F was supported in part by NIH predoctoral training grant T32GM007287. We thank the Bloomington *Drosophila* Stock Center (NIH P40OD018537), Kate O'Conner-Giles, Hugo Bellen, Gerry Rubin and Stephan Sigrist for *Drosophila* stocks, Kate O'Conner-Giles (Brown University) for help with CRISPR, Avi Rodal (Brandeis University) and Noreen Reist (Colorado State University) for providing antisera, Jan Melom (MIT) for generating the UAS-NHE-GFP transgenic line, Dina Volfson (MIT) for assistance with *Drosophila* stock generation, and members of the Littleton lab for helpful discussions and comments on the manuscript.

## Additional information

### Funding

| Funder | Grant reference number | Author |
| --- | --- | --- |
| National Institutes of Health | NS40296 | J Troy Littleton |
| National Institutes of Health | MH104536 | J Troy Littleton |
| National Institutes of Health | T32GM007287 | Monica C Quiñones-Frías |

The funders had no role in study design, data collection and interpretation, or the decision to submit the work for publication.

### Author contributions

Zhuo Guan, Conceptualization, Formal analysis, Investigation, Writing - original draft; Monica C Quiñones-Frías, Conceptualization, Data curation, Formal analysis, Investigation, Writing - original draft, Writing - review and editing; Yulia Akbergenova, Data curation, Formal analysis, Investigation; J Troy Littleton, Conceptualization, Formal analysis, Supervision, Funding acquisition, Validation, Writing - original draft, Project administration, Writing - review and editing

### Author ORCIDs

J Troy Littleton https://orcid.org/0000-0001-5576-2887

### Decision letter and Author response

Decision letter https://doi.org/10.7554/eLife.55443.sa1
Author response https://doi.org/10.7554/eLife.55443.sa2

## Additional files

### Supplementary files

• Transparent reporting form

### Data availability

All data generated during this study are included in the manuscript and supporting files.

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
