## [Decision Letter]

**Acceptance summary:**

Synaptotagmin 7 has been implicated as a calcium sensor involved in the short-term dynamics of neurotransmitter release at synapses throughout the mammalian central nervous system. This paper from the laboratory of Dr. Troy Littleton examines the function of Synaptotagmin 7 at a well characterized invertebrate synapse, the *Drosophila* NMJ, where high resolution quantification of neurotransmitter release characteristics can be combined with routine imaging and electron microscopy. The authors provide evidence that Synaptotagmin 7 negatively regulates baseline neurotransmitter release, a distinct function compared to existing studies, and an important addition to the field of synaptic vesicle fusion and synaptic transmission.

**Decision letter after peer review:**

Thank you for submitting your article "*Drosophila* Synaptotagmin 7 negatively regulates synaptic vesicle fusion and replenishment in a dosage-dependent manner" for consideration by *eLife*. Your article has been reviewed by three peer reviewers, one of whom is a member of our Board of Reviewing Editors, and the evaluation has been overseen by Gary Westbrook as the Senior Editor. The reviewers have opted to remain anonymous.

Summary

Guan et al. examine the role of Synaptotagmin 7 (*Syt7*) in synaptic transmission at the larval *Drosophila* NMJ. This is an interesting topic, and the field is currently focused on the possibility that *Syt7* is a calcium sensor responsible for the short-term dynamics of neurotransmitter release, although there are other models currently being debated in the literature. In the present study, electrophysiological analysis of newly generated *Syt7* null mutants revealed a pronounced (~2-fold) increase in neurotransmitter release in heterozygous and homozygous *Syt7* mutants. Conversely, neural *Syt7* overexpression decreased release. *Syt7* mutants display higher release probability, decreased short-term facilitation, increased RRP size and faster recovery from synaptic depression. Release of *Syt7*/*Syt1* double mutants was larger than in *Syt1* mutants alone, implying that *Syt7* negatively regulates release independent of *Syt1*. Changes in synaptic physiology did not correlate with apparent changes in synaptic morphology or ultrastructure. Endogenously tagged *Syt7* localizes to synaptic boutons, where it does not co-localize with several markers, including the active zone marker Bruchpilot. Based on these data, the authors propose a model in which *Syt7* negatively regulates the release and resupply of synaptic vesicles. In general, the study is based on a considerable amount of solid data and the major phenotype is interesting, as it implies a new role for *Syt7* in negative release regulation that differs from previous findings in different species.

Three reviewers with detailed knowledge of the mechanisms of neurotransmitter release and genetic analyses in model organisms examined and discussed the paper. There was considerable discussion about the major conclusions of the study. The reviewers generally agree on a set of major deficits. As the authors can see below, there was also considerable debate regarding more extensive experiments that would substantially strengthen the work, but which were ultimately deemed un-necessary at this time. Thus, the focus on the major experimental requirements reflects both a need for new data/analysis and careful consideration by the reviewers as to the experiments that are essential versus experiments that would simply extend impact. Regarding the major experiments requested:

First, there was discussion as to whether or not the current data fully refute existing models of *Syt7* function. In this regard, there is a notable absence of data to directly interrogate the calcium-dependence of neurotransmission. These experiments are straightforward and do not necessitate additional resources and would greatly strengthen the study, regardless of outcome.

Second, the reviewers discussed the need for genetic rescue. It is clear from the literature that tagged SYT proteins can interrupt normal trafficking. Given that tagged over-expression constructs are the major source of information for the peri-active zone model of *Syt7* function in *Drosophila*, additional genetic experiments are essential in order to demonstrate genetic rescue and substantiate the use of the tagged transgenes. Again, no new reagents are necessary, and the experiments are straightforward with time necessary only for several generations of crosses, which should take less than the 2-3 month interval in total and necessitate time investment at the end for analysis.

Third, there is a notable discrepancy with work previously published by the Littleton laboratory regarding *Syt7* function. In essence, an analysis was previously published for a mutation that was shown to eliminate *Syt7* function, and no functional phenotype was documented electrophysiologically. This discrepancy needs to be resolved. These and a few additional major issues are detailed below with additional specificity.

Major Issues:

1) As discussed by the authors, it remains completely elusive why loss of *Syt7* results in increased release. While this, in itself, is not necessarily a problem, there are missing experiments that are straightforward tests of the existing models that should be performed in order for this study to move the field forward sufficiently to justify publication in a general topic, high profile journal such as e*Life*. These include: A) test of calcium-dependence of release over a wide range of calcium concentrations (0.2-2mM) performed in voltage clamp. B) Direct measurement of presynaptic, action-potential induced calcium influx. This is the most labor-intensive experiment requested, but one that is deemed essential given the existing literature and given that *Syt7* is a calcium-binding protein. This experiment is well within the expertise of the Littleton laboratory given their elegant analyses of postsynaptic calcium dynamics. With respect to the necessary nature of these experiments, one reviewer wrote, "Of particular interest, Liu et al., 2014 (Chapman lab) concluded (using cultured hippocampal neurons from *Syt7* KO mice) that *Syt7* is the calcium sensor for SV replenishment during high frequency stimulation. Removal of *Syt7* had no effect on the basal RRP but significantly slowed recruitment of SVs to the RRP during high frequency stimulation and this function depended on its ability to bind calcium. This stands in direct contrast to the results shown here. Without the addition of clear data probing calcium influx presynaptically, and without additional data on the calcium-dependence of release and vesicle coupling, the present study fails to directly or adequately address the other major hypothesis in the field."

2) The authors put forward an interesting new model for *Syt7* function that is based on anatomical localization of the *Syt7* protein at the NMJ. In order to validate this model, it is essential that the tagged protein be shown to function correctly. Therefore, the authors should demonstrate that the tagged constructs achieve full rescue of the null mutation.

3) Related to point number 2, the authors should shore up the genetic argument that *Syt7* is dose dependent. The authors should include genetic rescue to confirm the nature of the mutations. Furthermore, the authors should place their putative null mutation in trans to a deletion of the gene locus to demonstrate, unequivocally, that the null is a clear loss of function mutation. This is particularly important given that release is enhanced in the loss of function mutant background.

4) Control EJC amplitudes in Figure 3 (showing the reduction in the OE condition) are equivalent to the experimental amplitudes in Figure 2 (reporting an increase in the mutant). In other words, the authors must explain why the controls are not consistent, and why the effects in one experiment are no different than the controls in another. If these are differences in genetic background, then this should be clearly stated and supported by the inclusion of additional data, possibly in a supplemental figure. Alternatively, there may be a need for additional experiments to clarify this discrepancy.

5) The authors need to come to directly address their own previously published data. In the (Sarawati et al., 2007) paper they demonstrate complete elimination of *Syt7* protein at the NMJ using RNAi. However, the EPSP amplitudes at 0.1 and 0.4mM external calcium are identical to controls. This is not the phenotype that they currently present. The field will not be able to make a judgment regarding the correct answer. The Littleton lab must have thought about this issue and is the only laboratory that is able to make sense of the discrepancy. The current paper should not be published without adequate resolution of this issue, either with clear new text added to the manuscript or with experimentation.

6) Figure 2A, 3A, and 7B: The noise of the average mEJC traces is (almost perfectly) correlated between genotypes. How were these average traces computed? How can (regular) noise be correlated between different recordings/experimental groups? Was a noise cancelling device utilized? If so, the authors need to address how this affects amplitude and kinetic measurements and state in methods. Alternative explanations?

7) The title implies that *Syt7* regulates synaptic fusion. However, the study does not provide direct evidence that *Syt7* negatively regulates the fusion process. I therefore suggest changing the title accordingly.

---

## [Author Response]

Major Issues:1) As discussed by the authors, it remains completely elusive why loss of Syt7 results in increased release. While this, in itself, is not necessarily a problem, there are missing experiments that are straightforward tests of the existing models that should be performed in order for this study to move the field forward sufficiently to justify publication in a general topic, high profile journal such as eLife. These include: A) test of calcium-dependence of release over a wide range of calcium concentrations (0.2-2mM) performed in voltage clamp. B) Direct measurement of presynaptic, action-potential induced calcium influx. This is the most labor-intensive experiment requested, but one that is deemed essential given the existing literature and given that Syt7 is a calcium-binding protein. This experiment is well within the expertise of the Littleton laboratory given their elegant analyses of postsynaptic calcium dynamics. With respect to the necessary nature of these experiments, one reviewer wrote, "Of particular interest, Liu et al., 2014 (Chapman lab) concluded (using cultured hippocampal neurons from Syt7 KO mice) that Syt7 is the calcium sensor for SV replenishment during high frequency stimulation. Removal of Syt7 had no effect on the basal RRP but significantly slowed recruitment of SVs to the RRP during high frequency stimulation and this function depended on its ability to bind calcium. This stands in direct contrast to the results shown here. Without the addition of clear data probing calcium influx presynaptically, and without additional data on the calcium-dependence of release and vesicle coupling, the present study fails to directly or adequately address the other major hypothesis in the field."

These are excellent suggestions for expanding our study and represent experiments that we had completed before the coronavirus-mandated closure of research labs by MIT. We present the results of these two major experiments in the revised manuscript. Together, they have allowed us to further define the mechanisms contributing to the enhanced evoked release observed in *Syt7* mutants.

For requested experiment A, we performed TEVC recordings in control and *Syt7^M1^*nulls in eight extracellular Ca^2+^concentrations: 0.175, 0.2, 0.3, 0.4, 0.6, 0.8, 1, and 2 mM. The slopes from a linear fit of a double logarithmic plot of evoked responses in the linear range was used to determine the Ca^2+^cooperativity of release. This data is now included as new panel L in Figure 2. *Syt7^M1^*mutants display enhanced evoked release over the entire range of extracellular Ca^2+^, shifting the curve leftward. Regression analysis revealed no significant difference in Ca^2+^cooperativity between control and *Syt7^M1^*, indicating the enhanced release observed is not due to alterations in the Ca^2+^cooperativity of release.

For requested experiment B, we used Fluo-4 AM to measure Ca^2+^ influx during short stimulation trains in *Syt7^M1^* mutants and controls. If SYT7 normally restricts SV release by acting as a Ca^2+^ buffer or reducing presynaptic Ca^2+^ influx, we expect to observe enhanced Ca^2+^ influx in *Syt7* mutants. In contrast, we observed significantly lower Ca^2+^ influx in *Syt7^M1^* compared to controls. We have included this data as new panels L and K in Figure 4. These data indicate SYT7 does not function as a Ca^2+^ buffer or a negative regulator of Ca^2+^ channel function to reduce release. Although it is unknown why Ca^2+^ influx is reduced in *Syt7* mutants, these data fit with our observation that BRP T-bar volume is reduced in *Syt7^M1^* as shown in Figure 4J. We suspect these changes may represent a homeostatic compensation to the enhanced release in Syt7 mutants as a mechanism to downregulate presynaptic output, indicating we may be underestimating the magnitude of enhanced release that occurs when SYT7 is absent.

2) The authors put forward an interesting new model for Syt7 function that is based on anatomical localization of the Syt7 protein at the NMJ. In order to validate this model, it is essential that the tagged protein be shown to function correctly. Therefore, the authors should demonstrate that the tagged constructs achieve full rescue of the null mutation.

The reviewers note the exciting new data on subcellular localization of SYT7 at peri-AZs that we presented. However, this tagged construct is not a UAS rescue line, which could result in mis-localization of SYT7 due to overexpression. Instead, it is a much better tool – a C-terminal CRISPR tag of endogenous SYT7, and therefore cannot be used for rescue. However, the reviewers’ request to show the tag does not inactivate SYT7 function is still important to verify. Indeed, we made extensive use of SYT1 C-terminal tags in our prior research of this isoform and have not observed any disruption of SYT1 function by C-terminal tagging. Similar work using FLASH-FALI tagging of *Drosophila* SYT1 has also been done and again this tagging did not disrupt SYT1 function in rescue experiments. As such, we expected C-terminal tagging of SYT7 would be tolerated by this isoform as well. To confirm the endogenous CRISPR RFP-tagged line we used for localization studies did not inactivate SYT7, we performed TEVC recordings on the control CRISPR injection strain and SYT7^RFP^. We observed no significant differences in the evoked response between the genotypes. These data have now been added to the Results section of the manuscript.

3) Related to point number 2, the authors should shore up the genetic argument that Syt7 is dose dependent. The authors should include genetic rescue to confirm the nature of the mutations. Furthermore, the authors should place their putative null mutation in trans to a deletion of the gene locus to demonstrate, unequivocally, that the null is a clear loss of function mutation. This is particularly important given that release is enhanced in the loss of function mutant background.

The extreme dosage sensitivity of SYT7 in both directions makes rescue difficult, as generating a precise match of the endogenous levels of SYT7 with GAL4 drivers is something we haven’t achieved. However, we provide extensive experimental data in the manuscript to support our hypothesis of dosage-sensitivity, including data from two independently generated Syt7 null mutants (CRISPR generated null and Minos insertion before C2A leading to a premature stop). Both of these mutants lack SYT7 protein by Western as shown in Figure 1E. Heterozygotes have an intermediate phenotype for multiple phenotypes we tested, including the enhanced evoked release (shown in Figure 2) and enhanced recovery of the RRP after 10 Hz stimulation (Figure 10). In addition, overexpression of UAS-*Syt7* with *elav^C155^*-GAL4 strongly suppresses release (Figure 3) and strongly suppresses SV recovery after 10 Hz stimulation (Figure 10—figure supplement 1). These data arise from multiple independently generated genetic manipulations and provide convincing support for our conclusion that SYT7 regulates neurotransmission in a dose-dependent manner.

4) Control EJC amplitudes in Figure 3 (showing the reduction in the OE condition) are equivalent to the experimental amplitudes in Figure 2 (reporting an increase in the mutant). In other words, the authors must explain why the controls are not consistent, and why the effects in one experiment are no different than the controls in another. If these are differences in genetic background, then this should be clearly stated and supported by the inclusion of additional data, possibly in a supplemental figure. Alternatively, there may be a need for additional experiments to clarify this discrepancy.

As the reviewers point out, differences in EJC amplitude can be seen in distinct genetic backgrounds. This is indeed what the reviewers are noticing in the EJC amplitudes in Figures 2 and 3, and why we always precisely match genetic backgrounds for all experiments in the lab. The larger responses in Figure 3 are due to the *elav^C155^*-GAL4 genetic background compared to the white CRISRP injection background in Figure 2. *elav^C155^*-GAL4 has larger evoked responses compared to white, and we have only made comparisons between precisely matched genetic backgrounds in each experiment described in the manuscript. The *elav^C155^*-GAL4 background is used as the appropriate control for *elav^C155^*-GAL4, UAS-*Syt7* overexpression. The control for our null mutant recordings is the precisely matched CRISPR injection line that was sequence verified.

5) The authors need to come to directly address their own previously published data. In the (Sarawati et al., 2007) paper they demonstrate complete elimination of Syt7 protein at the NMJ using RNAi. However, the EPSP amplitudes at 0.1 and 0.4mM external calcium are identical to controls. This is not the phenotype that they currently present. The field will not be able to make a judgment regarding the correct answer. The Littleton lab must have thought about this issue and is the only laboratory that is able to make sense of the discrepancy. The current paper should not be published without adequate resolution of this issue, either with clear new text added to the manuscript or with experimentation.

As noted by the reviewers, a previous RNAi approach to study SYT7 did not reveal an increase in evoked release. We did address this point in the Discussion in the prior version, but have expanded that section in the current resubmission. The original Syt7 RNAi was made by my group as a double stranded exon-intron hairpin at a time before RNAi libraries were available to the field. We have now compared the results of RNAi knockdown of the two commercial RNAi lines used in our current manuscript (labeled #1 and #2) with the previous line (labeled #3) from our prior publication. We drove expression of the 3 RNAi lines individually in the background of the endogenously-tagged SYT7^GFP^ CRISPR line and performed Western analysis. Indeed, the prior line (#3) we generated did not reduce endogenous SYT7 expression, allowing us to conclude that the prior RNAi was ineffective in reducing endogenous SYT7. We’ve expanded the Discussion of this result in the current version and have included a Western of the result in Author response image 1.

**Author response image 1. sa2fig1:** Three RNAis were tested for pan-neuronal knockdown of SYT7^GFP^. Western blot was probed with anti-GFP. Lane #1: UAS-*Syt7* RNAi#1 (Vienna #24989); SYT7^GFP^, lane #2: *elav^C155^*-GAL4, UAS-Dicer2; UAS-*Syt7* RNAi#1; SYT7^GFP^, lane #3: UAS-*Syt7* RNAi#2 (BDSC #27279); SYT7^GFP^, lane #4: *elav^C155^*-GAL4, UAS-Dicer2; UAS-*Syt7* RNAi#2, lane #5: *elav^C155^*-GAL4, UAS-Dicer2; SYT7^GFP^ and lane #6: *elav^C155^*-GAL4, UAS-Dicer2; UAS-*Syt7* RNAi#3 (Saraswati et al., 2007); SYT7^GFP^.

6) Figure 2A, 3A, and 7B: The noise of the average mEJC traces is (almost perfectly) correlated between genotypes. How were these average traces computed? How can (regular) noise be correlated between different recordings/experimental groups? Was a noise cancelling device utilized? If so, the authors need to address how this affects amplitude and kinetic measurements and state in methods. Alternative explanations?

Indeed, we attribute these effects to gaussian noise filtering of the mini traces by the amplifier. Mini amplitudes are close to background noise levels and filtering exaggerates noise corrections during the falling phase of the response as it bleeds into background from the rig. While the decay componentof the mini data is noisy, the main conclusions of the study are not affected given we are reporting mini frequency and amplitude. As mini amplitude is calculated from base to peak, noise in the decay phase does not affect amplitude measurements. Evoked responses are much larger, and no filtering is applied for these recordings.

7) The title implies that Syt7 regulates synaptic fusion. However, the study does not provide direct evidence that Syt7 negatively regulates the fusion process. I therefore suggest changing the title accordingly.

As suggested by the reviewer, we have changed the title to replace “fusion” with “release”.

The title now reads “*Drosophila* Synaptotagmin 7 negatively regulates synaptic vesicle release and replenishment in a dosage-dependent manner”.